# LEASE: OFFLINE PREFERENCE-BASED REINFORCEMENT LEARNING WITH HIGH SAMPLE EFFICIENCY

## ABSTRACT

Offline preference-based reinforcement learning (PbRL) provides an effective way to overcome the challenges of designing reward and the high costs of online interaction. However, since labeling preference needs real-time human feedback, acquiring sufficient preference labels is challenging. To solve this, this paper proposes a offLine prEference-bAsed RL with high Sample Efficiency (LEASE) algorithm, where a learned transition model is leveraged to generate unlabeled preference data. Considering the pretrained reward model may generate incorrect labels for unlabeled data, we design an uncertainty-aware mechanism to ensure the performance of reward model, where only high confidence and low variance data are selected. Moreover, we provide the generalization bound of reward model to analyze the factors influencing reward accuracy, and demonstrate that the policy learned by LEASE has theoretical improvement guarantee. The developed theory is based on state-action pair, which can be easily combined with other offline algorithms. The experimental results show that LEASE can achieve comparable performance to baseline under fewer preference data without online interaction.

## 1 INTRODUCTION

Reinforcement learning (RL) has been widely applied in robot control (Haarnoja et al., 2024; Radosavovic et al., 2024) and healthcare (Li et al., 2021; Wu et al., 2022) fields. However, in real-world scenarios, RL faces two serious challenges: 1) It is dangerous for agent to interact directly with environment, especially human-in-loop control (Levine et al., 2020); 2) It is difficult to design a quantitative reward function to accurately reflect the human intention or preference (Hadfield-Menell et al., 2017). Many researches have been explored to solve the above challenges.

Offline RL, learning policy from previous collected dataset without interaction with environment, has become an effective way for challenge 1 (Kumar et al., 2020; Kostrikov et al., 2021a). Preference-based RL (PbRL), which learns the reward function from human preference data without the demand for tedious hand-engineered reward design (Christiano et al., 2017), has been developed recently for challenge 2. However, compared with data $(s, a, s')$, the collecting cost of preference data is higher since it demands human real-time feedback to label preference, which often brings high sample complexity (low sample efficiency) (Liang et al., 2022; Park et al., 2022).

Some data augmentation techniques have been proposed to solve problem of the limitation of preference data (Park et al., 2022; Hu et al., 2024). However, the above methods need online interaction with environment and still demand large human feedback. Therefore, this paper aims to design a high sample efficiency offline PbRL algorithm that can achieve comparable control performance with baseline from a limited amount of preference data without interaction with environment.

In offline PbRL setting, existing methods (Shin et al., 2021; Kim et al., 2023; Zhang et al., 2024) typically involve two steps: reward learning and policy learning. However, the theoretical guarantees of reward generalization and policy improvement are not provided. Therefore, this paper focuses on answering the following two questions: 1) In algorithm, how to use the limited preference data to learn accurate reward model and guarantee agent performance? 2) In theory, what factors affect the reward generalization and what is the detailed relationship between offline RL algorithm itself, the reward gap and the improvement of policy performance?

Figure 1: The learning framework of LEASE. The offline dataset without reward is used to train transition model and limited labeled preference dataset is used to pretrain reward model. Then the generated unlabeled preference data is screened through uncertainty-aware selecting mechanism. The reward model is updated based on labeled and generated dataset. Finally, the agent learns the policy based on offline dataset and learned reward model.
.

For question 1, motivated by model-based offline RL, which learns the environment model to broaden data coverage (Yu et al., 2020; Liu et al., 2024), we also train the transition model to achieve data augmentation (improve sample efficiency). Specifically, the agent generates two different trajectories through interaction with the learned transition model, and then the pretrained reward model generates pseudo label for these two trajectories. However, the preference model may generate incorrect pseudo label, which may bring unstable training and low performance.

For question 2, there are very fewer algorithms for offline PbRL theory. The most relevant work is Zhan et al. (2024), where the theory of policy improvement guarantee is established. However, the generalization bound for reward model is not considered, and this theory is based on the whole trajectory rather than state-action pair in offline algorithms, which is not conducive to the theoretical analysis of specific offline PbRL algorithms since many offline RL are based on state-action pair.

To solve the above problems, this paper proposes a novel offLine prEference-bAsed RL with high Sample Efficiency algorithm (LEASE), where a selecting mechanism is designed to guarantee the quantity of generated dataset. Moreover, we develop a new generalization bound for reward model, and provide the theory of policy improvement guarantee for offline PbRL based on state-action pair. Fig. 1 shows the learning framework of LEASE. LEASE can achieve comparable performance to baseline under fewer preference data. The contributions are given below:

1) A novel learning framework, named LEASE, is proposed for offline PbRL, where an innovative preference augmentation technique is utilized to improve sample efficiency.

2) An uncertainty-aware mechanism is designed for screening the generated preference data so as to guarantee the stability of reward training and improve the accuracy of reward prediction.

3) The generalization bound of reward model and the theory of policy improvement based on state-action pair are developed to analyze factors that influence reward and policy performance.

In this paper, the theoretical and experimental results show that LEASE has policy improvement guarantee and can achieve superior performance under fewer preference data on D4RL benchmark.

## 2 PRELIMINARIES

**Offline Reinforcement Learning.** The framework of RL is based on the Markov Decision Process (MDP) that is described by the tuple $\mathcal{M} = (\mathcal{S}, \mathcal{A}, R, T, \rho, \gamma)$, where $\mathcal{S}$ is the state space, $\mathcal{A}$ is the action space, $R : \mathcal{S} \times \mathcal{A} \to \mathbb{R}$ is the reward function, $T : \mathcal{S} \times \mathcal{A} \to \Delta(\mathcal{S})$ is the transition dynamics, $\rho$ is the initial state distribution, and $\gamma \in (0, 1)$ is the discount factor (Levine et al., 2020). The term $\Delta(\Omega)$ denotes the set of probability distribution over space $\Omega$. The goal of RL is to optimize the policy $\pi$ that maximizes the expected discounted return $J(\pi, R) := \mathbb{E}_{(s,a) \sim d_T^\pi(s,a)}[R(s,a)]/(1-\gamma)$, where $d_T^\pi(s, a) := d_T^\pi(s)\pi(a|s)$ is the state-action marginal distribution under the learned policy $\pi$. The discounted state marginal distribution $d_T^\pi(s)$ is denoted as $(1 - \gamma)\sum_{t=0}^\infty \gamma^t \mathcal{P}(s_t = s|\pi)$,

where $\mathcal{P}(s_t = s|\pi)$ is the probability of reaching state $s$ at time $t$ by rolling out $\pi$. The policy $\pi$ can be derived from $Q$-learning, which learns the $Q$-function that satisfies Bellman operator: $\mathcal{T}Q(s, a) := R(s, a) + \gamma\mathbb{E}_{s'\sim T_\mathcal{M}(s'|s,a)}[\max_{a'\in\mathcal{A}} Q(s', a')]$ (Sutton et al., 1998).

The goal of offline RL is to learn a policy from offline dataset $\mathcal{D}_{\text{offline}} = \{(s_i, a_i, r_i, s_i')\}_{i=1}^C$ collected by the behavior policy $\mu$. The policy learning includes two parts: policy evaluation (minimizing the Bellman error) and policy improvement (maximizing the $Q$-function), that is

$$\widehat{Q} \leftarrow \arg\min_Q \mathbb{E}_{s,a,s'\sim\mathcal{D}}\big[Q(s, a) - \widehat{\mathcal{T}}\widehat{Q}(s, a)\big]^2 \qquad \text{(Policy Evaluation)}$$

$$\widehat{\pi} \leftarrow \arg\max_\pi \mathbb{E}_{s\sim\mathcal{D}, a\sim\pi}\big[\widehat{Q}(s, a)\big] \qquad \text{(Policy Improvement)} \tag{1}$$

where $\mathcal{D}$ can be a fixed offline dataset or replay buffer generated by the current policy $\widehat{\pi}$ interacting with the environment. The operator $\widehat{\mathcal{T}}$ is the Bellman operator based on sample, denoted as $\widehat{\mathcal{T}}Q(s, a) := R(s, a) + \gamma\max_{a'\in\mathcal{A}} Q(s', a')$ (Kumar et al., 2020).

**Preference-based Reinforcement Learning.** Different from standard RL setting, the reward function is not available in PbRL. Instead, PbRL learns the reward function $\widehat{R}$ from preferences between pairs of trajectory segments to align the human intention (Wilson et al., 2012), where a trajectory segment $\sigma$ of length $L$ is defined as a sequence of states and actions $\{s_k, a_k, ..., s_{k+L-1}, a_{k+L-1}\} \in (\mathcal{S} \times \mathcal{A})^L$. Given a pair of segments $(\sigma_0, \sigma_1)$, human choose which segment is preferred, i.e., $y \in \{0, 1\}$ (Christiano et al., 2017). The preference label $y = 1$ indicates $\sigma_1 \succ \sigma_0$ and $y = 0$ indicates $\sigma_0 \succ \sigma_1$, where $\sigma_i \succ \sigma_j$ denotes that the segment $i$ is preferable to the segment $j$.

The preference data is stored in the dataset $\mathcal{D}_l$ as a tuple $(\sigma_0, \sigma_1, y)$. To obtain the reward function $\widehat{R}$ parameterized by $\psi$, the Bradley-Terry model (Bradley and Terry, 1952) is utilized to define a preference predictor following previous works (Kim et al., 2023; Verma and Metcalf, 2024):

$$P(\sigma_0 \succ \sigma_1; \psi) = \frac{\exp\sum_t \widehat{R}_\psi(s_t^0, a_t^0)}{\exp\sum_t \widehat{R}_\psi(s_t^0, a_t^0) + \exp\sum_t \widehat{R}_\psi(s_t^1, a_t^1)}. \tag{2}$$

Then, based on the collected preference dataset $\mathcal{D}_l$, the reward function $\widehat{R}_\psi$ can be updated through minimizing the cross-entropy loss between the predicted and the true preference labels (Verma and Metcalf, 2024):

$$\mathcal{L}_R(\psi) = -\mathbb{E}_{(\sigma_0,\sigma_1,y)\sim\mathcal{D}_l}\big[(1-y)\log P(\sigma_0 \succ \sigma_1; \psi) + y\log P(\sigma_1 \succ \sigma_0; \psi)\big]. \tag{3}$$

After obtaining the reward function $\widehat{R}$, offline PbRL optimizes the policy by maximizing the expected discounted return $J(\pi, R)$ like standard offline RL.

**Model-based Offline Reinforcement Learning.** Model-based RL learns the dynamics model to improve sample efficiency (Rigter et al., 2022; Liu et al., 2024). They used the offline dataset $\mathcal{D}_{\text{offline}}$ to estimate transition model $\widehat{T}(s'|s, a)$. The transition model $\widehat{T}_\varphi$ parameterized by $\varphi$ is typically trained via maximum likelihood estimation (MLE):

$$\mathcal{L}_T(\varphi) = -\mathbb{E}_{(s,a,s')\sim\mathcal{D}_{\text{offline}}}\big[\log \widehat{T}_\varphi(s' \mid s, a)\big]. \tag{4}$$

In practical implantation, the transition model is approximated by Gaussian distribution and the MLE method is employed to train $N_T$ ensemble transition models $\{T_\varphi^i = \mathcal{N}(\mu_\varphi^i, \sigma_\varphi^i)\}_{i=1}^{N_T}$ (Yu et al., 2020). The samples are generated through $H$-step rollouts. Here, we use the trained transition to generate more trajectory to achieve data augmentation.

## 3 PROBLEM FORMULATION

The cost of collecting preference data is high since it needs real-time human feedback. Therefore, this paper aims to improve sample efficiency to learn accurate reward model from limited preference data and guarantee agent performance. For PbRL, the ideal form is that the learned reward function $\widehat{R}$ from collected preference data can be consistent with true reward $R^*$. Here, we define the function class as $\mathcal{R} = \{R : \mathcal{S} \times \mathcal{A} \to \mathbb{R}\}$. Then we assume the reward class $\mathcal{R}$ is realizable.

**Assumption 1** (Realizability). *Let $d(s, a) \in \Delta(\mathcal{S} \times \mathcal{A})$ be the arbitrary data distribution. Then, for any distribution $d(s, a)$, $\inf_{R\in\mathcal{R}} \mathbb{E}_{(s,a)\sim d(s,a)}[R^*(s, a) - R(s, a)]^2 < \varepsilon_R$ holds.*

Based on the above assumption, the optimal reward function can be obtained if the preference data can cover all data space. However, preference data tend to be limited, even less than hundred in real-world scenarios, such as rehabilitation robot field. Therefore, different from previous PbRL (Park et al., 2022; Hu et al., 2024; Zhang et al., 2024), this paper learns reward function from the limited dataset $\mathcal{D}_l = \{(\sigma_0^l, \sigma_1^l, y)^{(i)}\}_{i=1}^{N_l}$. We train the transition model through Eq. (4) to generate more unlabeled preference data $\mathcal{D}_u = \{(\sigma_0^u, \sigma_1^u)^{(i)}\}_{i=1}^{N_u}$. The pseudo labels $\widehat{y}$ for unlabeled dataset $\mathcal{D}_u$ can be obtained through reward model trained on $\mathcal{D}_l$ (Park et al., 2022), that is

$$\widehat{y}(\sigma_0^u, \sigma_1^u) = \mathbb{1}\big[P(\sigma_1^u \succ \sigma_0^u; \psi) > 0.5\big], \tag{5}$$

where $\mathbb{1}(\cdot)$ is indicator function. The reward function is updated through collected labeled dataset $\mathcal{D}_l$ and generated unlabeled dataset $\mathcal{D}_u$. Then, the reward model can be optimized through minimizing

$$\mathcal{L}'_R(\psi) = \mathbb{E}_{(\sigma_0^l, \sigma_1^l, y) \sim \mathcal{D}_l}\big[L\left((\sigma_0^l, \sigma_1^l), y\right)\big] + \mathbb{E}_{(\sigma_0^u, \sigma_1^u) \sim \mathcal{D}_u}\big[L\left((\sigma_0^u, \sigma_1^u), \widehat{y}\right)\big], \tag{6}$$

where $L((\sigma_0, \sigma_1), y) = -(1-y)\log P(\sigma_0 \succ \sigma_1; \psi) - y\log P(\sigma_1 \succ \sigma_0; \psi)$. However, the pretrained reward model may generate incorrect pseudo-labels for unlabeled dataset, leading to noisy training and poor generalization (Rizve et al., 2021). Therefore, one key question is how to design a data selecting mechanism to improve prediction accuracy and guarantee training stability.

The another key aspect is the theory for offline PbRL. There are very few algorithms specifically designed for offline PbRL with strong guarantee, including generalization bound for reward model and safe improvement guarantee for policy learning. Therefore, another key question is to develop a systematic theory for offline PbRL, including generalization bound and improvement guarantee.

## 4 REWARD LEARNING

The reward learning involves two stages: pretraining based on collected labeled data $\mathcal{D}_l$ and updating based on $\mathcal{D}_l$ and unlabeled data $\mathcal{D}_u$ during policy learning, which is essentially semi-supervised learning (Berthelot et al., 2019). This section focuses on designing a data selecting function $f(\sigma_0^u, \sigma_1^u)$ to ensure the quality of generated preference data, and explaining the factors that influence the generalization ability of reward model.

### 4.1 UNCERTAINTY-AWARE PSEUDO-LABEL SELECTION FOR REWARD LEARNING

Motivated by the previous pseudo-labeling work (Rizve et al., 2021), we select data from unlabeled dataset $\mathcal{D}_u$ according two principles: confidence and uncertainty. The data with high confidence and low uncertainty can be chosen for reward training. High confidence refers that pre-trained reward model discriminates the preference of two trajectories into $\widehat{y}$ with high probability $p(\sigma_0^u, \sigma_1^u, \widehat{y})$. Low uncertainty refers that the $N_R$ reward models (model ensembles) predicts with small variance $\tau(\sigma_0^u, \sigma_1^u, N_R)$. The probability confidence $p(\sigma_0^u, \sigma_1^u, \widehat{y})$ and uncertainty variance $\tau(\sigma_0^u, \sigma_1^u, N_R)$ can be denoted as

$$\begin{aligned} p(\sigma_0^u, \sigma_1^u, \widehat{y}) &= (1 - \widehat{y}) \cdot \bar{P}(\sigma_0^u \succ \sigma_1^u; \psi) + \widehat{y} \cdot \bar{P}(\sigma_1^u \succ \sigma_0^u; \psi) \\ \tau(\sigma_0^u, \sigma_1^u, N_R) &= \texttt{Std}\{\boldsymbol{P}(\sigma_0^u \succ \sigma_1^u; \psi)\}, \end{aligned} \tag{7}$$

where $N_R$ is the number of reward model, $\texttt{Std}\{\boldsymbol{P}(\cdot)\}$ denotes the variance of output probability of $N_R$ reward models, and $\bar{P}(\cdot)$ is the mean probability of $N_R$ reward models. Therefore, according to high confidence and low uncertainty, the $f(\sigma_0^u, \sigma_1^u)$ can be denoted as:

$$f(\sigma_0^u, \sigma_1^u) = \mathbb{1}\Big[p(\sigma_0^u, \sigma_1^u, \widehat{y}) > \kappa_p\Big] \cdot \mathbb{1}\Big[\tau(\sigma_0^u, \sigma_1^u, N_R) < \kappa_\tau\Big], \tag{8}$$

where $\kappa_p$ and $\kappa_\tau$ are thresholds of confidence and uncertainty respectively. $f(\sigma_0^u, \sigma_1^u) = 1$ denotes the generated data $(\sigma_0^u, \sigma_1^u)$ is selected, and $f(\sigma_0^u, \sigma_1^u) = 0$ denotes the data is not selected. Then combing Eq. (6), the reward model can be optimized through minimizing

$$\widehat{\mathcal{L}}_R(\psi) = \frac{1}{N_l}\sum_{i=1}^{N_l} L\left((\sigma_0^l, \sigma_1^l)^{(i)}, y^i\right) + \frac{1}{\widetilde{N}_u}\sum_{j=1}^{N_u} f\left((\sigma_0^u, \sigma_1^u)^{(j)}\right) \cdot L\left((\sigma_0^u, \sigma_1^u)^{(j)}, \widehat{y}^j\right), \tag{9}$$

where $\widetilde{N}_u$ is the number of generated dataset after screening, and $\widehat{\mathcal{L}}_R(\psi)$ is the empirical risk. It contains two parts: labeled loss $\widehat{\mathcal{L}}_l(\psi)$ and unlabeled loss $\widehat{\mathcal{L}}_u(\psi)$. Note that the label in unlabeled

loss is pseudo-label, which may be different from the true label. Therefore, there is the gap between the unlabeled loss with pseudo-label $\widehat{\mathcal{L}}_u(\psi)$ and that with true label $\widehat{\mathcal{L}'}_u(\psi)$. Before bounding this gap, we firstly give the below assumption without loss of generality.

**Assumption 2.** *For the pretrained reward model $\widehat{R}_\psi$ through the limited labeled dataset $\mathcal{D}_l = \{(\sigma_0^l, \sigma_1^l, y)^{(i)}\}_{i=1}^{N_l}$, if the pseudo label $\widehat{y}$ is defined in Eq. (5), then the pseudo-labeling error for the unlabeled dataset $\mathcal{D}_u = \{(\sigma_0^u, \sigma_1^u)^{(j)}\}_{j=1}^{N_u}$ is smaller than $\eta$, i.e., $\sum_{j=1}^{N_u} \mathbb{1}[\widehat{y}^j \neq y^j]/N_u \leq \eta$.*

**Proposition 1.** *Suppose that the loss $L((\sigma_0, \sigma_1), y))$ is bounded by $\Omega$. Then, for any $R \in \mathcal{R}$, under Assumption 2 the following equation holds:*

$$\left| \widehat{\mathcal{L}}_u(\psi) - \widehat{\mathcal{L}'}_u(\psi) \right| \leq \eta\Omega. \tag{10}$$

The proof of Proposition 1 can be found in Appendix A.1. This gap is mainly influenced by the term $\eta$. Through selecting high confidence and low uncertainty samples, the $\eta$ can be significantly reduced. Combining Eqs. (5) and (8), we find that if $p(\sigma_0^u, \sigma_1^u, \widehat{y})$ is small, the $L((\sigma_0^u, \sigma_1^u), \widehat{y})$ would become larger. Too large error can lead to unstable training and low performance. Therefore, through selecting mechanism $f(\sigma_0^u, \sigma_1^u)$, a more accurate subset of pseudo-labels can be used in reward training, and through setting a higher confidence threshold for pseudo labeling, a lower prediction error and the stability of training can be guaranteed.

### 4.2 GENERALIZATION BOUND FOR REWARD MODEL

Before developing generalization bound, we define the expected error $\mathcal{L}_R(\psi)$ with respect to the reward model $R(s, a; \psi)$: $\mathcal{L}_R(\psi) = \mathbb{E}_{(\sigma_0, \sigma_1, y) \sim \mathcal{D}}[L((\sigma_0, \sigma_1), y)]$, where $\mathcal{D}$ is data distribution. The reward model is trained through minimizing the empirical error $\widehat{\mathcal{L}}_R(\psi)$ in Eq. (9). Through developing generalization error bound, we can analyse the factors that influence generalization ability.

Similar to the previous generalization bound works (Xie et al., 2024), Rademacher complexity, which measures the richness of a certain hypothesis space (Mohri and Muñoz Medina, 2012), is introduced firstly. The definition is given below:

**Definition 1** (Empirical Rademacher complexity)**.** *Let $\mathcal{F}$ be a family of functions mapping from $\mathcal{Z}$ to $\mathbb{R}$ and $\widehat{\mathcal{S}} = \{z_1, \ldots, z_{N_s}\}$ be a fixed sample of size $N_s$ drawn from the distribution $\mathcal{S}$ over $\mathcal{Z}$. The empirical Rademacher complexity of $\mathcal{G}$ for sample $\widehat{\mathcal{S}}$ is defined as*

$$\widehat{\mathfrak{R}}_{\widehat{\mathcal{S}}}(\mathcal{F}) = \mathbb{E}_{\boldsymbol{\sigma}} \Big[ \sup_{f \in \mathcal{F}} \frac{1}{N_s} \sum_{i=1}^{N_s} \sigma_i f(z_i) \Big], \tag{11}$$

*where $\boldsymbol{\sigma} = (\sigma_1, ..., \sigma_N)$ are independent uniform random variables taking values in $\{-1, +1\}$.*

**Definition 2** (Induced Reward Function Families)**.** *Given for a space $\mathcal{R}$ of reward function $R$, the induced reward function families $\Pi(\mathcal{F})$ is defined as*

$$\Pi(\mathcal{R}) = \big\{ (\sigma_0, \sigma_1) \to -\log P(\sigma_0 \succ \sigma_1) \mid R \in \mathcal{R} \big\}, \tag{12}$$

*where $P(\cdot)$ is defined in Eq. (2) and $\Pi(\mathcal{R})$ is the union of projections of $\mathcal{R}$ onto each dimension.*

In general, lower Rademacher complexity corresponds to better generalization performance. Then, based on the above definition and Proposition 1, we develop the new generalization bound for reward model trained through Eq. (9). The specific theorem is given below.

**Theorem 1.** *Let reward model be trained on the labeled dataset $\mathcal{D}_l = \{(\sigma_0^l, \sigma_1^l, y)^{(i)}\}_{i=1}^{N_l}$ and unlabeled dataset $\mathcal{D}_u = \{(\sigma_0^u, \sigma_1^u)^{(i)}\}_{i=1}^{N_u}$. Then, for any $\delta > 0$, with probability at least $1 - \delta$, under Assumption 1 and 2 the following holds for any reward function $R \in \mathcal{R}$,*

$$\mathcal{L}_R(\psi) \leq \widehat{\mathcal{L}}_R(\psi) + \eta\Omega + 4\widehat{\mathfrak{R}}_{\widehat{\mathcal{D}}}\big(\Pi(\mathcal{R})\big) + 3\sqrt{\frac{\log(2/\delta)}{2(N_l + N_u)}}, \tag{13}$$

*where $\widehat{\mathcal{D}}$ is the input combination of labeled and unlabeled dataset, i.e. $\{(\sigma_0, \sigma_1)^{(i)}\}_{i=1}^{N_l+N_u}$.*

---

**Algorithm 1** LEASE: off**L**ine pr**E**ference-b**A**sed RL with high **S**ample **E**fficiency

---

**Input**: offline dataset $\mathcal{D}_{\text{offline}}$, limited labeled preference dataset $\mathcal{D}_l$, critic $Q_\omega$, policy $\pi_\theta$, transition model $T_\varphi$, and reward model $R_\psi$.

**Output**: the reward model $\widehat{R}$ and the policy $\widehat{\pi}$ learned by LEASE.

1: **Initialization:** Randomly initializing all networks and generated unlabeled dataset $\mathcal{D}_u = \varnothing$.
2: Train an ensemble transition model $\{T_\varphi^i\}_{i=1}^{N_T}$ on $\mathcal{D}_{\text{offline}}$ according to Eq. (4).
3: Pre-train an ensemble reward model $\{R_\psi^i\}_{i=1}^{N_R}$ on $\mathcal{D}_l$ according to Eq. (3).
4: **for** $t = 1, 2, \cdots, N_{\text{iter}}$ **do**
5:     **if** $t \ \% \ rollout\ frequency = 0$ **then**
6:         Generate synthetic $H$-step rollouts by $T_\varphi$. Add transition trajectory into $\mathcal{D}_u$.
7:     **end if**
8:     **if** *reward update condition* is TRUE **then**
9:         Give pseudo label for unlabeled dataset $\mathcal{D}_u$ based on pre-trained reward model.
10:        Select high confidence and low uncertainty unlabeled data according to Eq. (8).
11:        Train the reward model based on $\mathcal{D}_l$ and $\mathcal{D}_u$ according to Eq. (9).
12:    **end if**
13:    Train policy using offline RL algorithms, such as CQL and IQL.
14: **end for**

---

The proof of Theorem 1 can be found in Appendix A.2. Theorem 1 indicates that the expected error $\mathcal{L}_R(\psi)$ is bounded by empirical error $\widehat{\mathcal{L}}_R(\psi)$, pseudo-labeling error $\eta\Omega$, Rademacher complexity and constant terms. The constant term can become small when the number of unlabeled data increases, but it may cause unstable training when labels are inaccurate. According to Proposition 1, empirical error and pseudo-labeling error can be reduced through selecting mechanism $f(\sigma_0^u, \sigma_1^u)$, thus the better generalization ability, that is tighter upper bound, can be achieved. Note that Theorem 1 has generality and can be applicable to methods that train reward model using pseudo-labels. The difference among various methods may lie in how they train reward model to improve the accuracy of pseudo-labels, that is reducing $\eta$. If one method fails to reduce the pseudo-labeling error $\eta$, the upper generalization bound would be looser than that of no data augmentation.

## 5 POLICY LEARNING

This paper directly uses the proposed offline RL algorithms, such as CQL (Kumar et al., 2020) and IQL (Kostrikov et al., 2021b), to perform policy learning. To solve the problem of high cost for labeling preference, we propose a off**L**ine pr**E**ference-b**A**sed RL with high **S**ample **E**fficiency algorithm (LEASE). This section aims to describe the details of LEASE and establishes the theory for safe policy improvement guarantee.

### 5.1 THE IMPLEMENTATION DETAILS OF LEASE

Most offline RL algorithms are based on Eq. (1). They proposed various algorithms to solve the problem of distribution shift. In offline PbRL setting, the reward is unknown for agent. The accuracy of reward model directly influence the performance of policy. Moreover, the unlabeled data are generated through the current policy interaction with learned transition model. Therefore, the reward model and policy influence each other. In practical implementation, the policy, $Q$-function, reward and transition model are parameterized by $\pi_\theta$, $Q_\omega$, $R_\psi$ and $T_\varphi$ respectively.

Algorithm 1 gives pseudocode for LEASE. The goal of LEASE is to learn better reward model $R_\psi$ from fewer labeled preference dataset $\mathcal{D}_l$ and learn better policy $\pi_\theta$ from offline dataset $\mathcal{D}_{\text{offline}}$ without interaction with environment. The key of LEASE is how to train reward model in policy learning process. Notably, the reward model only update once instead of updating constantly in this process. The reward update condition is set to when the number of unlabeled data in buffer reaches the maximum buffer capacity. The update time is influenced by rollout frequency and rollout batch size. The reward update condition, rollout length $H$ and selecting mechanism $f(\sigma_1, \sigma_2)$ are the important factors for reward model. The more detailed analysis can be found in Appendix B.3.

## 5.2 SAFE POLICY IMPROVEMENT GUARANTEE

Offline RL aims to guarantee $\xi$-safe policy improvement over the behavior policy $\mu$ (the policy used to collect offline dataset), *i.e.* $J(\widehat{\pi}, R^*) \geq J(\mu, R^*) - \xi$ (Kumar et al., 2020). This part develops theoretical guarantee of policy improvement for LEASE, that is giving the bound for $J(\mu, R^*) - J(\widehat{\pi}, R^*)$, and further analyzes sample complexity when reward model is unknown. Firstly, we propose a new single-policy concentrability coefficient for PbRL based on state-action pair instead of trajectory like (Zhan et al., 2024).

**Definition 3** (Concentrability coefficient for PbRL). *Concentrability coefficient $\mathscr{C}_\mathcal{R}(\pi)$ is used to measure how well reward model errors transfer between offline data distribution $d_T^\mu$ and the visitation distribution $d_T^\pi$ under transition $T$ and policy $\pi$, defined as*

$$\mathscr{C}_\mathcal{R}(\pi) = \sup_{\widehat{R} \in \mathcal{R}} \left| \frac{\mathbb{E}_{(s,a) \sim d_T^\pi} \left[ R^*(s,a) - \widehat{R}(s,a) \right]}{\mathbb{E}_{(s,a) \sim d_T^\mu} \left[ R^*(s,a) - \widehat{R}(s,a) \right]} \right|. \tag{14}$$

*where $R^*(s,a)$ is the true reward model and coefficient $\mathscr{C}_\mathcal{R}(\pi)$ is upper bounded by $\|d_T^\pi/d_T^\mu\|_\infty$.*

Next, similar to (Geer, 2000; Zhan et al., 2024), we use $\varepsilon$-bracketing number to measure the complexity of reward function class $\mathcal{R}$, which can be defined as

**Definition 4** ($\varepsilon$-bracketing number). *The $\varepsilon$-bracketing number $\mathcal{N}_\mathcal{R}(\varepsilon)$ is the minimum number of $\varepsilon$-brackets $(l, u)$ required to cover a function class $\mathcal{R}$, where each bracket $(l, u)$ satisfies $l(\sigma_0, \sigma_1) \leq P_R(\sigma_0, \sigma_1) \leq u(\sigma_0, \sigma_1)$ and $\|l - u\|_1 \leq \varepsilon$ for all $R \in \mathcal{R}$ and all trajectory-pairs $(\sigma_0, \sigma_1)$, and $P_R(\sigma_0, \sigma_1)$ is the probability that segment $\sigma_0$ is preferable to segment $\sigma_1$ defined in Eq. (2).*

**Theorem 2.** *Under Assumption 1, for any $\delta \in (0, 1]$, the policy $\widehat{\pi}$ learned by LEASE, with high probability $1 - \delta$, satisfies that $J(\mu, R^*) - J(\widehat{\pi}, R^*)$ is upper bound by*

$$\xi + \frac{1 + \mathscr{C}_\mathcal{R}(\widehat{\pi})}{1 - \gamma} \left( \sqrt{\frac{4C}{NL^2} \log \left( \frac{\mathcal{N}_\mathcal{R}(1/N)}{\delta} \right)} + \sqrt{\frac{4R_{max}^2 \log (1/\delta)}{NL}} \right), \tag{15}$$

*where $C > 0$ is the absolute constant defined in Eq. (A.19), $N$ is the size of preference dataset, and $L$ is the length of trajectory. The term $\xi$ is performance gap depending on offline algorithm itself.*

The proof of Theorem 2 can be found in Appendix A.3. The $\xi$ is constant when offline algorithm is determined. Therefore, the tighter bound can be achieved through reducing performance gap caused by reward model (see Proposition 3). It can be reduced to small value $\varpi$ with sample complexity of

$$N = \widetilde{\mathcal{O}} \left( \frac{4(1 + \mathscr{C}_\mathcal{R}(\widehat{\pi}))^2}{\varpi^2(1 - \gamma)^2} \left( \sqrt{\frac{C \log(\mathcal{N}_\mathcal{R}(1/N)/\delta)}{L^2}} + \sqrt{\frac{R_{max}^2 \log (1/\delta)}{L}} \right)^2 \right), \tag{16}$$

where concentrability coefficient $\mathscr{C}_\mathcal{R}(\widehat{\pi})$ can become smaller when the learned policy $\widehat{\pi}$ is close to behavior policy $\mu$. Bracketing number $\log(\mathcal{N}_\mathcal{R}(1/N))$ measures the complexity of function class $\mathcal{R}$ and takes $\widetilde{\mathcal{O}}(d)$ in linear reward model (Zhan et al., 2024). Notably, LEASE can learn accurate reward model through data augmentation under fewer $N_l$ preference data ($N_l < N$). Thus, the performance gap can become tighter under fewer data, that is LEASE can have higher sample efficiency compared with Zhan et al. (2024). More discussion can be found in Appendix C.1.

## 6 EXPERIMENTS

This section focuses on answering the following questions: **Q₁**: How well the accuracy and generalization of reward model through data augmentation under fewer preference data? (Section 6.2) **Q₂**: How well does LEASE perform compared with offline PbRL baseline in standard benchmark tasks? (Section 6.1) We answer these questions using D4RL benchmark (Fu et al., 2020) with several control tasks. More hyperparameters and implementation details are provided in Appendix B. The code for LEASE is available at github.com/***.

Table 1: Results for the D4RL tasks during the last 5 iterations of training averaged over 3 seeds. ± captures the standard deviation over seeds. Bold indicates the performance within 2% of the best performing algorithm for each offline algorithm.

| Task Name | CQL (Kumar et al., 2020) | | | IQL (Kostrikov et al., 2021b) | | |
|---|---|---|---|---|---|---|
| | URLHF | FEWER | LEASE | URLHF | FEWER | LEASE |
| walker2d-m | $76.0 \pm 0.9$ | $77.4 \pm 0.6$ | $\mathbf{78.4 \pm 0.9}$ | $\mathbf{78.4 \pm 0.5}$ | $71.8 \pm 0.8$ | $74.6 \pm 1.8$ |
| walker2d-m-e | $92.8 \pm 22.4$ | $77.7 \pm 0.3$ | $\mathbf{98.6 \pm 18.1}$ | $\mathbf{109.4 \pm 0.1}$ | $105.2 \pm 3.1$ | $\mathbf{108.1 \pm 0.5}$ |
| hopper-m | $54.7 \pm 3.4$ | $55.8 \pm 2.8$ | $\mathbf{56.5 \pm 0.6}$ | $50.8 \pm 4.2$ | $\mathbf{56.7 \pm 2.6}$ | $\mathbf{56.0 \pm 0.5}$ |
| hopper-m-e | $\mathbf{57.4 \pm 4.9}$ | $53.6 \pm 0.9$ | $56.4 \pm 0.8$ | $\mathbf{94.3 \pm 7.4}$ | $56.0 \pm 1.4$ | $55.9 \pm 1.9$ |
| halfcheetah-m | $\mathbf{43.4 \pm 0.1}$ | $\mathbf{43.5 \pm 0.1}$ | $\mathbf{43.5 \pm 0.1}$ | $\mathbf{43.3 \pm 0.2}$ | $42.2 \pm 0.1$ | $\mathbf{43.0 \pm 0.3}$ |
| halfcheetah-m-e | $\mathbf{62.7 \pm 7.1}$ | $48.3 \pm 0.7$ | $53.2 \pm 3.1$ | $\mathbf{91.0 \pm 2.3}$ | $59.1 \pm 4.9$ | $62.4 \pm 1.4$ |
| **Mujoco Average** | $\mathbf{64.5 \pm 6.5}$ | $59.4 \pm 0.9$ | $\mathbf{64.4 \pm 4.0}$ | $\mathbf{77.9 \pm 2.5}$ | $65.2 \pm 2.2$ | $66.7 \pm 1.0$ |
| pen-human | $\mathbf{9.8 \pm 14.1}$ | $0.5 \pm 3.0$ | $3.8 \pm 4.6$ | $50.2 \pm 15.8$ | $67.3 \pm 10.0$ | $\mathbf{75.6 \pm 3.3}$ |
| pen-expert | $\mathbf{138.3 \pm 5.2}$ | $128.1 \pm 0.7$ | $132.5 \pm 2.3$ | $\mathbf{132.9 \pm 4.6}$ | $104.1 \pm 12.9$ | $113.8 \pm 6.3$ |
| door-human | $\mathbf{4.7 \pm 5.9}$ | $0.2 \pm 1.0$ | $\mathbf{4.7 \pm 8.8}$ | $3.5 \pm 3.2$ | $4.0 \pm 2.5$ | $\mathbf{5.9 \pm 0.5}$ |
| door-expert | $\mathbf{103.9 \pm 0.8}$ | $103.0 \pm 0.9$ | $\mathbf{103.2 \pm 0.7}$ | $\mathbf{105.4 \pm 0.4}$ | $104.5 \pm 0.6$ | $\mathbf{105.2 \pm 0.2}$ |
| hammer-human | $\mathbf{0.9 \pm 0.3}$ | $0.3 \pm 0.0$ | $0.3 \pm 0.0$ | $1.4 \pm 1.0$ | $1.2 \pm 0.2$ | $\mathbf{1.7 \pm 0.4}$ |
| hammer-expert | $120.2 \pm 6.8$ | $124.1 \pm 2.1$ | $\mathbf{126.3 \pm 1.2}$ | $\mathbf{127.4 \pm 0.2}$ | $125.2 \pm 2.3$ | $\mathbf{126.3 \pm 0.1}$ |
| **Adroit Average** | $\mathbf{63.0 \pm 5.5}$ | $59.4 \pm 1.3$ | $61.8 \pm 3.0$ | $70.1 \pm 4.2$ | $67.1 \pm 4.8$ | $\mathbf{71.4 \pm 1.8}$ |

## 6.1 RESULTS ON BENCHMARK TASKS

This part chooses several mujoco and adroit tasks to evaluate the performance of LEASE, where offline dataset and preference dataset originate from Fu et al. (2020) and Yuan et al. (2024), respectively. Yuan et al. (2024) provided baseline for offline PbRL based on 2000 preference data labeled by two types (We denote this method as URLHF). One is crowd-sourced labels obtained by crowd-sourcing, and the other is synthetic labels based on ground truth reward. To test LEASE performance for two types preference data, the labels of preference is from human-realistic feedback for mujoco tasks, and ground truth reward for adroit tasks. This paper aims to test the performance under fewer preference data without online interaction. We take the first 100 data of (Yuan et al., 2024) as 2000 preference dataset, and denote method without data augmentation as FEWER. The effects of the number of labeled preference data $N_l$ can be found in Appendix B.3.

Table 2: The comparison results between the performance using selecting mechanism and that not using. The latter method is denoted as FRESH. ↑ denotes the improvement of performance.

| Task Name | | walker2d-m | hopper-m | halfcheetah-m | walker2d-m-e | hopper-m-e | halfcheetah-m-e |
|---|---|---|---|---|---|---|---|
| CQL | FRESH | $76.0 \pm 1.3$ | $54.4 \pm 1.6$ | $43.4 \pm 0.1$ | $77.4 \pm 2.2$ | $55.0 \pm 0.4$ | $49.9 \pm 0.9$ |
| | LEASE | $\mathbf{78.4} \uparrow \mathbf{(3.3\%)}$ | $\mathbf{56.5} \uparrow \mathbf{(4.0\%)}$ | $\mathbf{43.5} \uparrow \mathbf{(0.6\%)}$ | $\mathbf{98.6} \uparrow \mathbf{(27.4\%)}$ | $\mathbf{56.4} \uparrow \mathbf{(2.6\%)}$ | $\mathbf{53.2} \uparrow \mathbf{(6.6\%)}$ |
| IQL | FRESH | $72.7 \pm 4.1$ | $53.5 \pm 0.8$ | $42.5 \pm 0.1$ | $105.3 \pm 1.0$ | $53.9 \pm 0.1$ | $54.5 \pm 2.2$ |
| | LEASE | $\mathbf{74.6} \uparrow \mathbf{(2.6\%)}$ | $\mathbf{56.0} \uparrow \mathbf{(4.5\%)}$ | $\mathbf{43.0} \uparrow \mathbf{(1.3\%)}$ | $\mathbf{108.1} \uparrow \mathbf{(2.7\%)}$ | $\mathbf{55.9} \uparrow \mathbf{(3.7\%)}$ | $\mathbf{62.4} \uparrow \mathbf{(14.6\%)}$ |

Here, we evaluate LEASE performance based on CQL (Kumar et al., 2020) and IQL (Kostrikov et al., 2021b) two offline algorithms. Table 1 gives the results of the average normalized score with standard deviation, where m and m-e denote medium and medium-expert respectively. The offline PbRL algorithm under sufficient feedback (URLHF) are obtained from paper (Yuan et al., 2024). This table shows that LEASE can greatly improve performance under fewer preference data compared with FEWER, and achieve comparable performance to URLHF that demand large human feedback. This validates the content of Theorem 2 that LEASE can reduce the performance gap caused by reward model under fewer preference data and has high sample efficiency. Table 2 shows the effect of designed selecting mechanism for agent performance, where the method not using selecting mechanism is denoted as FRESH. This table indicates that the application of selecting mechanism $f(\sigma_0, \sigma_1)$ can efficiently improve agent performance. The comparison results between LEASE and URLHF under fewer preference data, and results of model-based offline RL algorithm under the designed framework can be found in Appendix B.4.

## 6.2 RESULTS FOR REWARD MODEL

Learning an accurate reward is difficult in the offline PbRL setting when preference labels are scarce. Fig. 2 illustrates the relationship between the prediction accuracy of preference model without data augmentation and the number of preference data $N_l$, where preference model is based on reward

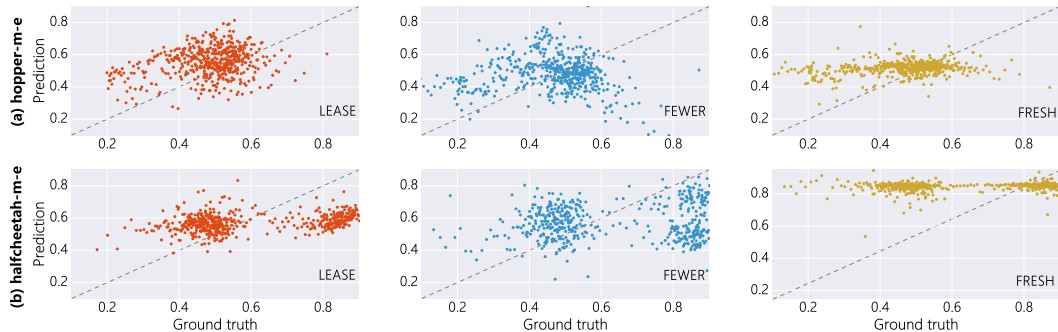

Figure 3: The comparison between prediction value by the learned rewards and their ground truths for different methods under (a) hopper-medium-expert and (b) halfcheetah-medium-expert datasets, where the value predicted by the trained reward model and ground-truth reward value are both normalized to $[0, 1]$. From left to right are methods LEASE, FEWER and FRESH, respectively.

.

Table 3: The performance of the learned transition model, where error is calculated by the sum of the mean square values of predicted and true value in each dimension.

| Task Name | walker2d-m | walker2d-m-e | hopper-m | hopper-m-e | halfcheetah-m | halfcheetah-m-e |
|---|---|---|---|---|---|---|
| Error | $10.19 \pm 0.30$ | $6.57 \pm 0.10$ | $0.35 \pm 0.01$ | $0.38 \pm 0.01$ | $16.18 \pm 0.21$ | $16.19 \pm 0.48$ |

| Task Name | pen-human | pen-expert | door-human | door-expert | hammer-human | hammer-expert |
|---|---|---|---|---|---|---|
| Error | $1.02 \pm 0.08$ | $1.05 \pm 0.01$ | $0.06 \pm 0.04$ | $0.04 \pm 0.01$ | $0.21 \pm 0.05$ | $0.58 \pm 0.03$ |

model (Eq. (2)). Here, we use the last 200 of 2000 preference data that are not seen in training stage as evaluation dataset. It indicates that as the number of preference data $N_l$ increases, the prediction accuracy of the preference model shows an upward trend. This also validates Theorem 1 that increasing the number of samples $N_l$ can reduce the generalization error.

Next, we compare the accuracy of reward model before and after data augmentation under fewer preference dataset. Fig. 3 shows the comparison results between prediction and ground truth of reward, where the predicted and true rewards are both normalized to $[0, 1]$. We randomly sample 500 data from unlabeled datasets that are not seen in training stage for evaluation. This figure indicates the linear relationship between reward predicted by LEASE and ground truth is better than that of other two methods. The prediction of reward model learned by method FRESH is very narrow and the accuracy is greatly reduced compared with FEWER. This is because generated data of FRESH are not selected by $f(\sigma_0, \sigma_1)$, which causes substantial errors for the labels of generated data and leads to the collapse of training. This validates that the performance of reward model can be improved through data augmentation and selecting mechanism, where the designed selecting mechanism has a greater impact on reward model performance. The more related experimental results can be found in Appendix B.4.

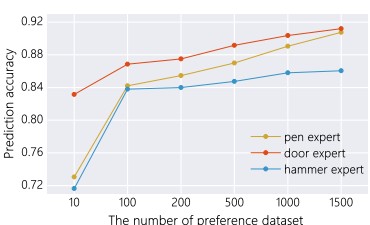

Figure 2: The relationship between preference model accuracy and the number of preference dataset $N_l$.

### 6.3 RESULTS FOR TRANSITION MODEL

The accuracy of transition model is the key factor influencing agent performance. Table 3 shows the accuracy result for the learned transition model, where the accuracy of transition model is measured by the sum of the mean square values of the predicted value and the true value in each dimension. Note that, for walker2d and halfcheetah tasks, the error of the learned transition model is greatly larger than hopper tasks in mujoco environment. This is mainly because the state space of walker2d and hopper is larger than hopper, and the physical model of them are more complex. Intuitively, the lower the accuracy of the transition model, the poorer performance of the agent. The results of the accuracy of the learned transition model for agent performance can be found in Appendix B.4.

## 7 RELATED WORK

**The algorithm for offline PbRL.** Offline PbRL eliminates the demand for interaction with environment and handcraft designed reward. OPAL (Shin et al., 2021) is the first algorithm that combining offline RL and PbRL. PT (Kim et al., 2023) utilized transformer-based architecture to design preference model capable of generating non-Markovian rewards. OPPO (Kang et al., 2023) directly optimized the policy in a high-level embedding space without learning a separate reward function. However, the above methods demands a large amount of preference dataset. Zhang et al. (2024) trains a diffusion model to achieve data augmentation, but this consumers larger training cost. Our method `LEASE` can achieve superior performance under fewer dataset and time.

**The theory for offline PbRL.** There are few algorithms that provide theoretical guarantees for offline PbRL, including the generalization bound of reward model and the guarantee of policy improvement. Zhu et al. (2023) studied offline PbRL, but the analysis are restricted to linear model. Zhan et al. (2024) extended to general function approximation, but the generalization analysis of reward model is not provided and the theory is based on trajectory. `LEASE` gives the theoretical analysis for reward model, and provides the theoretical guarantee for policy based on state-action pair. The theory of `LEASE` can be easily combined with other offline RL theory.

**Semi-supervised learning.** The goal of semi-supervised learning is using unlabeled data to improve model performance when labeled data is limited. Data selection techniques is used to filter the data with clean labels from a noisy dataset. Han et al. (2018) selected unlabeled data with small losses from one network. Li et al. (2020) used a two-component GMM to separate the dataset into a clean set and a noisy set. Rizve et al. (2021) modeled the prediction uncertainty of unlabeled data to screen data. Xiao et al. (2023) selected data with high confidence as clean data. Motivated by (Rizve et al., 2021) and (Xiao et al., 2023), we trained ensemble reward model to select data with high confidence and low variance to guarantee the quality of unlabeled data. The more related works for this study can be found in Appendix C.2.

## 8 CONCLUSION

This paper proposes a novel offline PbRL algorithm (`LEASE`) with high sample efficiency. `LEASE` can achieve comparable performance under fewer preference dataset. By selecting high confidence and low variance data, the stability and accuracy of reward model are guaranteed. Moreover, this paper provides the theoretical analysis for `LEASE`, including generalization of reward model and policy improvement guarantee. This theory can be easily connected with other offline algorithms. The theoretical and experimental results demonstrate that the data selecting mechanism $f(\sigma_0, \sigma_1)$ can effectively improve performance of reward model and the performance learned by `LEASE` can be guaranteed under fewer preference dataset. However, there is still the gap between the true and the learned reward model. Future works can focus on how to further reduce this gap under the limited preference data or how to achieve conservative estimation for state-action pairs where the learned reward model predicts inaccurately.

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

# A  RELATED PROOFS

This section gives the detailed proof for the theorems in the main text.

## A.1  PROOF OF PROPOSITION 1

**The Proof for Proposition 1**: This proof follows the previous work (Xie et al., 2024). Obviously, the largest pseudo-labeling error $\eta$ in Assumption 2 includes exactly two types of pseudo-labeling error:

$$\eta_1 = \frac{1}{N_u} \sum_{j=1}^{N_u} \mathbb{1}\Big[P\left(\sigma_1^j \succ \sigma_0^j; \psi\right) > 0.5, y^j = 0\Big],$$

$$\eta_2 = \frac{1}{N_u} \sum_{j=1}^{N_u} \mathbb{1}\Big[P\left(\sigma_0^j \succ \sigma_1^j; \psi\right) > 0.5, y^j = 1\Big], \tag{A.1}$$

where $\eta_1$ represents the error ratio of classifying the first category as the second category, and $\eta_2$ is the error ratio of classifying the second category as the first category. Then, we prove the gap between the unlabeled loss with pseudo-label $\widehat{\mathcal{L}}_u(\psi)$ and that with true label $\widehat{\mathcal{L}}'_u(\psi)$ from two sides: $\widehat{\mathcal{L}}_u(\psi) \le \widehat{\mathcal{L}}'_u(\psi) + \eta\Omega$ and $\widehat{\mathcal{L}}_u(\psi) \ge \widehat{\mathcal{L}}'_u(\psi) - \eta\Omega$. Notably, we ignore the selecting mechanism in the below proof since it doesn't influence the proof result.

**Step 1:** Proving upper bound : $\widehat{\mathcal{L}}_u(\psi) \le \widehat{\mathcal{L}}'_u(\psi) + \eta\Omega$.

$$\begin{aligned}
\widehat{\mathcal{L}}_u(\psi) =& \frac{1}{N_u} \sum_{j=1}^{N_u} L\left((\sigma_0^u, \sigma_1^u)^{(j)}, \widehat{y}^j\right) \\
=& -\frac{1}{N_u} \sum_{j=1}^{N_u} \mathbb{1}\big[P\left(\sigma_0^j \succ \sigma_1^j; \psi\right) > 0.5\big] \log P(\sigma_0^j \succ \sigma_1^j; \psi) \\
& + \mathbb{1}\big[P\left(\sigma_1^j \succ \sigma_0^j; \psi\right) > 0.5\big] \log P(\sigma_1^j \succ \sigma_0^j; \psi) \\
\le& -\frac{1}{N_u} \sum_{j=1}^{N_u} \mathbb{1}\big[y^j = 0\big] \log P(\sigma_0^j \succ \sigma_1^j; \psi) + \mathbb{1}\big[y^j = 1\big] \log P(\sigma_1^j \succ \sigma_0^j; \psi) \\
& + \mathbb{1}\Big[P\left(\sigma_1^j \succ \sigma_0^j; \psi\right) > 0.5, y^j = 0\Big] \log P(\sigma_0^j \succ \sigma_1^j; \psi) \\
& + \mathbb{1}\Big[P\left(\sigma_0^j \succ \sigma_1^j; \psi\right) > 0.5, y^j = 1\Big] \log P(\sigma_1^j \succ \sigma_0^j; \psi) \\
\le& \widehat{\mathcal{L}}'_u(\psi) + \eta_1 \max_j\Big\{-\log P(\sigma_0^j \succ \sigma_1^j)\Big\} + \eta_2 \max_j\Big\{-\log P(\sigma_1^j \succ \sigma_0^j)\Big\} \\
\le& \widehat{\mathcal{L}}'_u(\psi) + \eta\Omega.
\end{aligned} \tag{A.2}$$

**Step 2:** Proving low bound : $\widehat{\mathcal{L}}_u(\psi) \ge \widehat{\mathcal{L}}'_u(\psi) - \eta\Omega$.

$$\begin{aligned}
\widehat{\mathcal{L}}_u(\psi) =& \frac{1}{N_u} \sum_{j=1}^{N_u} L\left((\sigma_0^u, \sigma_1^u)^{(j)}, \widehat{y}^j\right) \\
\ge& -\frac{1}{N_u} \sum_{j=1}^{N_u} \mathbb{1}\big[y^j = 0\big] \log P(\sigma_0^j \succ \sigma_1^j; \psi) + \mathbb{1}\big[y^j = 1\big] \log P(\sigma_1^j \succ \sigma_0^j; \psi) \\
& - \mathbb{1}\Big[P\left(\sigma_1^j \succ \sigma_0^j; \psi\right) > 0.5, y^j = 0\Big] \log P(\sigma_0^j \succ \sigma_1^j; \psi) \\
& - \mathbb{1}\Big[P\left(\sigma_0^j \succ \sigma_1^j; \psi\right) > 0.5, y^j = 1\Big] \log P(\sigma_1^j \succ \sigma_0^j; \psi) \\
\ge& \widehat{\mathcal{L}}'_u(\psi) - \eta\Omega.
\end{aligned} \tag{A.3}$$

Combing Step 1 and Step 2, we can obtain the following result:

$$\left| \widehat{\mathcal{L}}_u(\psi) - \widehat{\mathcal{L}}'_u(\psi) \right| \leq \eta\Omega. \tag{A.4}$$

This completes the proof of Theorem 1.

## A.2 PROOF OF THEOREM 1

**Lemma 1.** *Let $\mathcal{F}$ be a family of functions mapping from $\mathcal{X}$ to $\mathbb{R}$ and $\widehat{\mathcal{D}}$ be empirical datasets sampled from an i.i.d. sample $\mathcal{D}$ of size $N$. Then, for any $\delta > 0$, with probability at least $1 - \delta$, the following holds for all $f \in \mathcal{F}$,*

$$\left| \mathbb{E}_{\mathcal{D}}\left[f\right] - \mathbb{E}_{\widehat{\mathcal{D}}}\left[f\right] \right| \leq 2\widehat{\mathfrak{R}}_{\widehat{\mathcal{D}}}(\mathcal{F}) + 3\sqrt{\frac{\log(2/\delta)}{2N}}, \tag{A.5}$$

*where $\mathbb{E}_{\widehat{\mathcal{D}}}\left[f\right] = \sum_{i=1}^{m}\left[g(x_i)\right]/m$ is the empirical form of $\mathbb{E}_{\mathcal{D}}\left[f\right]$. This proof can be found in Theorem 3.3 of work* Mohri et al. (2018).

**Proposition 2.** *Let $\mathcal{F}$ be a family of loss functions defined in Eq. (6) and $\Pi(\mathcal{R})$ be a family of functions defined in Definition 2. Then, for any sample $\widehat{\mathcal{D}} = \{(\sigma_0, \sigma_1, y)^{(i)}\}_{i=1}^{N}$, the following relation holds between the empirical Rademacher complexities of $\Pi(\mathcal{R})$ and $\mathcal{F}$:*

$$\widehat{\mathfrak{R}}_{\widehat{\mathcal{D}}}(\mathcal{F}) \leq 2\widehat{\mathfrak{R}}_{\widehat{\mathcal{D}}}\big(\Pi(\mathcal{R})\big). \tag{A.6}$$

***Proof.*** According to Definition 1, the empirical Rademacher complexity of $\mathcal{F}$ can be written as:

$$\begin{aligned}
&\widehat{\mathfrak{R}}_{\widehat{\mathcal{D}}}(\mathcal{F}) \\
=&\mathbb{E}_{\boldsymbol{\sigma}}\left[\sup_{R \in \mathcal{R}} \frac{1}{N}\sum_{i=1}^{N}\sigma_i\big[-(1-y)\log P\left(\sigma_0 \succ \sigma_1\right) - y\log P\left(\sigma_1 \succ \sigma_0\right)\big]\right] \\
\leq&\mathbb{E}_{\boldsymbol{\sigma}}\left[\sup_{R \in \mathcal{R}} \frac{1}{N}\sum_{i=1}^{N}\sigma_i\big[-\log P\left(\sigma_0 \succ \sigma_1\right)\big]\right] \\
&+ \mathbb{E}_{\boldsymbol{\sigma}}\left[\sup_{R \in \mathcal{R}} \frac{1}{N}\sum_{i=1}^{N}\sigma_i\big[-\log P\left(\sigma_1 \succ \sigma_0\right)\big]\right] \\
=&2\widehat{\mathfrak{R}}_{\widehat{\mathcal{D}}}\big(\Pi(\mathcal{R})\big).
\end{aligned} \tag{A.7}$$

This completes the proof of Proposition 2.

**The Proof for Theorem 1**: This proof is based on Proposition 1 and 2. Combing Eqs. (A.5) and (A.6), we can derive that for labeled dataset $\widehat{\mathcal{D}} = \{(\sigma_0, \sigma_1, y)^{(i)}\}_{i=1}^{N}$, the generalization error bound between expected error $\mathcal{L}(\psi)$ and empirical error $\widehat{\mathcal{L}}(\psi)$ holds:

$$\mathcal{L}(\psi) \leq \widehat{\mathcal{L}}(\psi) + 4\widehat{\mathfrak{R}}_{\widehat{\mathcal{D}}}\big(\Pi(\mathcal{R})\big) + 3\sqrt{\frac{\log(2/\delta)}{2N}}. \tag{A.8}$$

However, the dataset for training reward model includes two parts: the labeled dataset $\mathcal{D}_l = \{(\sigma_0^l, \sigma_1^l, y)^{(i)}\}_{i=1}^{N_l}$ and unlabeled dataset $\mathcal{D}_u = \{(\sigma_0^u, \sigma_1^u)^{(i)}\}_{i=1}^{N_u}$, and empirical error $\widehat{\mathcal{L}}_R(\psi) = \widehat{\mathcal{L}}_l(\psi) + \widehat{\mathcal{L}}_u(\psi)$. Let $\widehat{\mathcal{L}}'_u(\psi)$ be the empirical error under unlabeled dataset with true label, then

$$\mathcal{L}_R(\psi) \leq \widehat{\mathcal{L}}_l(\psi) + \widehat{\mathcal{L}}'_u(\psi) + 4\widehat{\mathfrak{R}}_{\widehat{\mathcal{D}}}\big(\Pi(\mathcal{R})\big) + 3\sqrt{\frac{\log(2/\delta)}{2(N_l + N_u)}}, \tag{A.9}$$

where $\widehat{\mathcal{D}}$ is the input combination of labeled and unlabeled dataset, denoted as $\{(\sigma_0, \sigma_1)^{(i)}\}_{i=1}^{N_l + N_u}$. According to Proposition 1, that is $\widehat{\mathcal{L}}'_u(\psi) \leq \widehat{\mathcal{L}}_u(\psi) + \eta\Omega.$, we can derive

$$\mathcal{L}_R(\psi) \leq \widehat{\mathcal{L}}_R(\psi) + \eta\Omega + 4\widehat{\mathfrak{R}}_{\widehat{\mathcal{D}}}\big(\Pi(\mathcal{R})\big) + 3\sqrt{\frac{\log(2/\delta)}{2(N_l + N_u)}}. \tag{A.10}$$

This completes the proof of Theorem 1.

### A.3 PROOF OF THEOREM 2

Before proof, we give the related Assumption and Lemmas for Theorem 2. Firstly, we assume the the reward class $\mathcal{R}$ is realizable (Assumption 1) and bounded (Assumption 3).

**Assumption 3** (Boundedness). *For any $R \in \mathcal{R}$ and any state-action pairs $(s, a)$, the equation $|R(s, a)| \leq R_{max}$ holds.*

**Proposition 3.** *Consider a set of trajectories $\{\sigma_0^i, \sigma_1^i\}_{i=1}^N$, each of length L, collected from an offline dataset following the distribution $d_T^\mu(s, a)$. Then, for any $\delta \in (0, 1]$, with probability at least $1 - \delta$, the following holds for all $\widehat{R} \in \mathcal{R}$*

$$\left| \mathbb{E}_{(s,a) \sim d_T^\mu(s,a)} \left[ R^*(s, a) - \widehat{R}(s, a) \right] \right| \leq \sqrt{\frac{4C}{NL^2} \log \left( \frac{\mathcal{N}_{\mathcal{R}}(1/N)}{\delta} \right)} + \sqrt{\frac{4R_{max}^2 \log (1/\delta)}{NL}}, \quad (A.11)$$

*where $R^*$ is the true reward model and $\widehat{R}$ is the learned reward model. $\mathcal{N}_{\mathcal{R}}(1/N)$ is the bracketing number defined in Definition 3.*

The proof of Proposition 3 can be found in Appendix A.4. Note that, the above reward model is only trained from offline preference dataset. Through generating more preference data, the upper bound can be tighter. However, the generated data may be inaccuracy, which may improve the reward gap. The selecting mechanism can effectively solve this problem.

**The Proof for Theorem 2**: This Theorem aims to give the lower bound of term $J(\widehat{\pi}, R^*) - J(\mu, R^*)$. We prove this from ideal case and actual case.

**1) Ideal case:** there is no need to consider distribution shift and empirical error problem. The policy $\widehat{\pi}$ is directly learned by $\widehat{\pi} = \max_\pi J(\pi, \widehat{R})$. Thus, $J(\widehat{\pi}, \widehat{R}) \geq J(\mu, \widehat{R})$ holds. Then, we can derive

$$\begin{aligned} J(\mu, R^*) - J(\widehat{\pi}, R^*) &= J(\mu, R^*) - J(\mu, \widehat{R}) + J(\mu, \widehat{R}) - J(\widehat{\pi}, R^*) \\ &\leq J(\mu, R^*) - J(\mu, \widehat{R}) + J(\widehat{\pi}, \widehat{R}) - J(\widehat{\pi}, R^*) \\ &\leq \left| J(\mu, R^*) - J(\mu, \widehat{R}) \right| + \left| J(\widehat{\pi}, R^*) - J(\widehat{\pi}, \widehat{R}) \right|. \end{aligned} \quad (A.12)$$

Since $J(\pi, R) := \mathbb{E}_{(s,a) \sim d_T^\pi(s,a)} [R(s, a)] / (1 - \gamma)$, $|J(\pi, R^*) - J(\pi, \widehat{R})|$ can be written as

$$\left| J(\pi, R^*) - J(\pi, \widehat{R}) \right| = \frac{1}{1 - \gamma} \left| \mathbb{E}_{(s,a) \sim d_T^\pi} \left[ R^*(s, a) - \widehat{R}(s, a) \right] \right|. \quad (A.13)$$

Then, according to Definition 3, we have

$$\left| J(\pi, R^*) - J(\pi, \widehat{R}) \right| \leq \frac{\mathscr{C}_{\mathcal{R}}(\pi)}{1 - \gamma} \left| \mathbb{E}_{(s,a) \sim d_T^\pi} \left[ R^*(s, a) - \widehat{R}(s, a) \right] \right|. \quad (A.14)$$

By Proposition 3, the following inequality holds

$$J(\mu, R^*) - J(\widehat{\pi}, R^*) \leq \frac{1 + \mathscr{C}_{\mathcal{R}}(\widehat{\pi})}{1 - \gamma} \left( \sqrt{\frac{4C}{NL^2} \log \left( \frac{\mathcal{N}_{\mathcal{R}}(1/N)}{\delta} \right)} + \sqrt{\frac{4R_{max}^2 \log (1/\delta)}{NL}} \right). \quad (A.15)$$

**2) Actual case:** offline RL suffers from distribution shift problem and it is a necessity to consider empirical error. Therefore, many offline RL algorithms incorporate conservatism into policy to overcome distribution shift, that is learning policy through $\widehat{\pi} = \max_\pi J(\pi, \widehat{R}) - P(\pi)$. Therefore, $J(\widehat{\pi}, \widehat{R}) \geq J(\mu, \widehat{R}) - \xi$ instead of $J(\widehat{\pi}, \widehat{R}) \geq J(\mu, \widehat{R})$ in actual implantation. The term $\xi$ depends on the algorithm itself. Here, we take CQL as a example. The policy gap is proven in Theorem 3.6 of (Kumar et al., 2020). The detailed is given as follow.

$$\xi = \underbrace{\frac{\gamma C_{T,R} R_{max}}{(1 - \gamma)^2} \mathbb{E}_{s \sim d_T^\mu(s)} \sqrt{\frac{|\mathcal{A}|(1 + D(s))}{|\mathcal{D}(s)|}}}_{:= \xi_1} - \underbrace{\frac{\alpha}{1 - \gamma} \mathbb{E}_{s \sim d_T^\mu(s)} \left[ D(s) \right]}_{:= \xi_2}, \quad (A.16)$$

where $| \cdot |$ denotes cardinality of a specific set, $D(s) = \sum_a \widehat{\pi}(a|s) \left( \frac{\widehat{\pi}(a|s)}{\mu(a|s)} - 1 \right)$ and $C_{T,R}$ is the empirical coefficient. It consists of two terms: the first term $\xi_1$ captures the decrease in policy

performance due to sampling error. The second term $\xi_2$ captures the increase in policy performance in empirical setting (Kumar et al., 2020).

Therefore, when considering distribution shift and empirical error, we can derive

$$
\begin{aligned}
J(\mu, R^*) - J(\widehat{\pi}, R^*) &\leq J(\mu, R^*) - J(\mu, \widehat{R}) + J(\widehat{\pi}, \widehat{R}) + \xi - J(\widehat{\pi}, R^*) \\
&\leq \xi + \left| J(\mu, R^*) - J(\mu, \widehat{R}) \right| + \left| J(\widehat{\pi}, R^*) - J(\widehat{\pi}, \widehat{R}) \right|.
\end{aligned}
\tag{A.17}
$$

Furthermore, according to Eq. (A.15), we have

$$
J(\mu, R^*) - J(\widehat{\pi}, R^*) \leq \xi + \frac{1 + \mathscr{C}_{\mathcal{R}}(\widehat{\pi})}{1 - \gamma} \left( \sqrt{\frac{4C}{NL^2} \log\left(\frac{\mathcal{N}_{\mathcal{R}}(1/N)}{\delta}\right)} + \sqrt{\frac{4R_{max}^2 \log(1/\delta)}{NL}} \right).
\tag{A.18}
$$

This completes the proof for Theorem 2.

## A.4 PROOF OF PROPOSITION 3

**Lemma 2.** *There exists an absolute constant $C$ such that for any $\delta \in (0, 1]$, with probability at least $1 - \delta$, the following holds for all $\widehat{R} \in \mathcal{R}$*

$$
\sum_{i=1}^{N} \left( P_{\widehat{R}}(y^i | \sigma_0^i, \sigma_1^i) - P_{R^*}(y^i | \sigma_0^i, \sigma_1^i) \right)^2 \leq C \log\left(\frac{\mathcal{N}_{\mathcal{R}}(1/N)}{\delta}\right),
\tag{A.19}
$$

*where $y \in \{0, 1\}$, $R^*$ is the true reward model and $\widehat{R}$ is the learned reward model. $P_R(0 \mid \sigma_0, \sigma_1)$ is the probability that $\sigma_0$ is preferable $\sigma_1$ under reward model $R$. $\mathcal{N}_{\mathcal{R}}(1/N)$ is the bracketing number defined in Definition 3. This proof can be found in Lemma 2 of previous work (Zhan et al., 2024) and Proposition 14 of (Liu et al., 2022).*

**The Proof for Proposition 3**: We prove this Proposition from two steps. Firstly, we bound the difference between $P_{R^*}(\cdot \mid \sigma_0, \sigma_1)$ and $P_{\widehat{R}}(\cdot \mid \sigma_0, \sigma_1)$. Then, we bound the difference between $R^*(s, a)$ and $\widehat{R}(s, a)$.

**Step 1:** Bound the probability difference $|P_{R^*}(\cdot \mid \sigma_0, \sigma_1) - P_{\widehat{R}}(\cdot \mid \sigma_0, \sigma_1)|$.

By Cauchy-Schwarz inequality $\left(\sum_i a_i b_i\right)^2 \leq \left(\sum_i a_i^2\right)\left(\sum_i b_i^2\right)$, we set $a_i = P_{\widehat{R}} - P_{R^*}$ and $b_i$ that is chosen from $\{-1, 1\}$ and satisfies $a_i b_i > 0$. Then, we have

$$
\frac{1}{N} \left( \sum_{i=1}^{N} \left| P_{\widehat{R}}(y^i | \sigma_0^i, \sigma_1^i) - P_{R^*}(y^i | \sigma_0^i, \sigma_1^i) \right| \right)^2 \leq \sum_{i=1}^{N} \left( P_{\widehat{R}}(y^i | \sigma_0^i, \sigma_1^i) - P_{R^*}(y^i | \sigma_0^i, \sigma_1^i) \right)^2.
$$

Then, by Lemma 2, the probability difference can be written as

$$
\sum_{i=1}^{N} \left| P_{\widehat{R}}(y^i | \sigma_0^i, \sigma_1^i) - P_{R^*}(y^i | \sigma_0^i, \sigma_1^i) \right| \leq \sqrt{CN \log\left(\frac{\mathcal{N}_{\mathcal{R}}(1/N)}{\delta}\right)}.
\tag{A.20}
$$

**Step 2:** Bound the reward difference $|R^*(s, a) - \widehat{R}(s, a)|$.

Based on Assumption 3, let $f(\sigma) = \sum_i R^*(s^i, a^i) \in [-LR_{max}, LR_{max}]$, $g(\sigma) = \sum_i \widehat{R}(s^i, a^i) \in [-LR_{max}, LR_{max}]$, and $F(x_1, x_2) = e^{x_1}/(e^{x_1} + e^{x_2})$. Then, according to Eq. (2), we can derive

$$
\left| P_{\widehat{R}}(y | \sigma_0, \sigma_1) - P_{R^*}(y | \sigma_0, \sigma_1) \right| = \left| F\big(f(\sigma_0), f(\sigma_1)\big) - F\big(g(\sigma_0), g(\sigma_1)\big) \right|.
\tag{A.21}
$$

Notably, the above equation only considers the condition $y = 0$ since the result of $y = 1$ is similar to $y = 0$. Then, according to error propagation, we have

$$
\begin{aligned}
& \left| F\big(f(\sigma_0), f(\sigma_1)\big) - F\big(g(\sigma_0), g(\sigma_1)\big) \right| \\
& \approx \left| \frac{\partial F}{\partial x_1} \right| \left| f(\sigma_0) - g(\sigma_0) \right| + \left| \frac{\partial F}{\partial x_2} \right| \left| f(\sigma_1) - g(\sigma_1) \right| \\
& \geq \frac{e^{x_1 + x_2}}{\left(e^{x_1} + e^{x_2}\right)^2} \left| \Big(f(\sigma_0) - g(\sigma_0)\Big) + \Big(f(\sigma_1) - g(\sigma_1)\Big) \right| \\
& \geq \frac{1}{4} \left| \sum_{l=1}^{L} \sum_{j=0}^{1} \Big( R^*(s_j^l, a_j^l) - \widehat{R}(s_j^l, a_j^l) \Big) \right|.
\end{aligned}
\tag{A.22}
$$

Then, combing Eqs. (A.20) and (A.22), we can further derive

$$
\left| \sum_{i=1}^{N} \sum_{l=1}^{L} \sum_{j=0}^{1} \Big( R^*(s_j^{i,l}, a_j^{i,l}) - \widehat{R}(s_j^{i,l}, a_j^{i,l}) \Big) \right| \leq 4 \sqrt{C N \log\left( \frac{\mathcal{N}_\mathcal{R}(1/N)}{\delta} \right)}.
\tag{A.23}
$$

According to Chernoff-Hoeffding bound (Hoeffding, 1994), for independent random variables $X_1, X_2, ..., X_n$, with high probability $1 - \delta$, the below equation holds

$$
\mathbb{E}[X] \leq \frac{1}{n} \sum_{i=1}^{n} X_i + (b - a) \sqrt{\frac{\log(1/\delta)}{2n}},
\tag{A.24}
$$

where $[a, b]$ is the range of values that each $X_i$ can take. Then, for Eq. (equation A.23), we set $X_i$ as $R^*(s_j, a_j) - R(s_j, a_j)$ and $X_i \in [-2R_{max}, 2R_{max}]$). Then, the below equation holds

$$
\left| \mathbb{E}_{(s,a) \sim d_T^\mu(s,a)} \big[ R^*(s, a) - \widehat{R}(s, a) \big] \right| \leq \sqrt{ \frac{4C}{NL^2} \log\left( \frac{\mathcal{N}_\mathcal{R}(1/N)}{\delta} \right) } + \sqrt{ \frac{4R_{max}^2 \log\left(1/\delta\right)}{NL} },
\tag{A.25}
$$

where $d_T^\mu(s, a)$ is the distribution of offline dataset. The above equation holds since the trajectories $\{\sigma_0^i, \sigma_1^i\}_{i=1}^{N}$ are collected from offline dataset. This completes the proof of Proposition 3.

## B  RELATED EXPERIMENTS

We conduct experiments on Mujoco and Adroit tasks, which are included in the D4RL (Fu et al., 2020) benchmark. The code for LEASE is available at github.com/***. This part introduces detailed experiments setup, hyper-parameter and parameter analysis for LEASE.

### B.1  EXPERIMENTS SETUP

**Offline dataset.** We employ Mujoco and Adroit environments to test the performance of LEASE. The Mujoco tasks include halfcheetah-v2, hopper-v2, and walker2d-v2. The Adroit tasks include pen-v1, door-v1 and hammer-v1. Fig. A.1 depicts the above six tasks. For three Mujoco tasks, the goal is to control the robot's various joints to achieve faster and more stable locomotion. For Adroit tasks, they involve controlling a 24-DoF simulated Shadow Hand robot on a sparse reward, high-dimensional robotic manipulation task (Fu et al., 2020). The offline datasets are based on the D4RL dataset. We select medium and medium-expert two types dataset for Mujoco tasks and human and expert for Adroit tasks. The difference between different dataset in certain task lies in the collected policy.

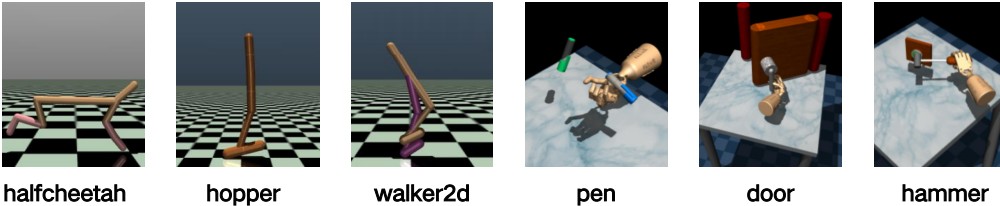

halfcheetah    hopper    walker2d    pen    door    hammer

Figure A.1: The description of Mujoco tasks (version 2) and Adroit tasks (version 1).
.

**Preference dataset.** The preference dataset is from (Yuan et al., 2024). We use two types preference dataset to test algorithm performance. One is that the labels for preference are from human feedback, the another is the labels are from the ground-truth reward. Obviously, since human preferences are subjective, the labels from ground-truth reward do not correspond to human preference in actual situation, and the performance of preference labels calibrated by ground-truth reward is better than that of labels calibrated by human feedback in most cases. Here, we use the labels from human feedback in Mujoco tasks and from ground-truth reward in Adroit tasks.

**Basic offline algorithms.** Offline PbRL setting involves two steps: reward learning and policy learning. Here, we directly use offline RL algorithms CQL (Kumar et al., 2020) and IQL (Kostrikov et al., 2021b) to perform policy learning. The performance of these algorithms is evaluated based on the cumulative return. For comparison, the scores are normalized between 0 (random policy score) and 100 (expert policy score) (Fu et al., 2020). The normalized score $\tilde{S}$ is computed by:

$$\tilde{S} = \frac{S - S_r}{S_e - S_r} \times 100,$$

where $S_r$, $S_e$ and $S$ are the expected return of a random policy, an expert policy, and the trained policy by offline RL algorithms, respectively.

### B.2 HYPER-PARAMETERS

**Reward and transition model training.** Table A.1 gives the hyperparameter configuration for reward and transition training. Similar to (Yu et al., 2020), we train an ensemble of 7 transition models and select the best 5 models. Each model consists of a 4-layer feed-forward neural network with 200 hidden units. The model training employs maximum likelihood estimation with a learning rate of $1e^{-3}$ and Adam optimizer. Following (Yuan et al., 2024), we train an ensemble of 3 reward models. Each reward model includes a 3-layer feed-forward neural network with 256 hidden units. The reward model training uses cross-entropy loss with a learning rate of $3e^{-4}$ and Adam optimizer.

Table A.1: Base parameter configuration of `LEASE`.

| Model | Parameter | Value |
|---|---|---|
| Transition Model | Model learning rate | $1 \times e^{-3}$ |
| | Number of hidden layers | 4 |
| | Number of hidden units per layer | 200 |
| | Batch size | 256 |
| | Number of model networks $N_T$ | 7 |
| | Number of elites | 5 |
| Reward Model | Model learning rate | $3 \times e^{-4}$ |
| | Number of hidden layers | 3 |
| | Number of hidden units per layer | 256 |
| | Batch size (Pretrain) | 256(64) |
| | Number of model networks $N_R$ | 3 |
| | Number of labeled dataset $N_l$ | 100 |

Table A.2: The hyperparameters for `LEASE` under CQL and IQL algorithms.

| Task Name | CQL | | | IQL | | |
|---|---|---|---|---|---|---|
| | $H$ | $\kappa_p$ | $\kappa_\tau$ | $H$ | $\kappa_p$ | $\kappa_\tau$ |
| walker2d-medium | 10 | 0.85 | 0.05 | 10 | 0.75 | 0.08 |
| walker2d-medium-expert | 10 | 0.85 | 0.05 | 10 | 0.85 | 0.08 |
| hopper-medium | 10 | 0.85 | 0.05 | 10 | 0.85 | 0.08 |
| hopper-medium-expert | 100 | 0.85 | 0.05 | 10 | 0.85 | 0.08 |
| halfcheetah-medium | 10 | 0.85 | 0.05 | 10 | 0.85 | 0.08 |
| halfcheetah-medium-expert | 200 | 0.85 | 0.05 | 10 | 0.75 | 0.08 |
| pen-human | 10 | 0.95 | 0.05 | 10 | 0.85 | 0.08 |
| pen-expert | 10 | 0.90 | 0.05 | 10 | 0.95 | 0.08 |
| door-human | 10 | 0.95 | 0.05 | 10 | 0.90 | 0.08 |
| door-expert | 20 | 0.95 | 0.05 | 10 | 0.95 | 0.08 |
| hammer-human | 10 | 0.95 | 0.05 | 10 | 0.90 | 0.08 |
| hammer-expert | 20 | 0.99 | 0.05 | 10 | 0.95 | 0.08 |

**Policy optimization.** Offline PbRL involves reward learning and policy optimization. For `LEASE`, the reward is updated during policy learning. Therefore, the reward is influenced by the rollout length and rollout batch size and current learned policy. Here, we select CQL and IQL as basic offline algorithms. In CQL algorithm, `cql weight` and `temperature` are set as 5.0 and 1.0 for all tasks, respectively. In IQL algorithm, `expectile` and `temperature` are set as 0.7 and 3.0 for mujoco tasks and 0.8 and 3.0 for adroit tasks.

The critic network $Q_\omega$ and the policy network $\pi_\theta$ adopt a 2-layer feed-forward neural network with 256 hidden units. The hyperparameters for `LEASE` includes: rollout horizon $H \in \{10, 20, 100, 200\}$, probability confidence $\kappa_p(\sigma_0^u, \sigma_1^u, \widehat{y}) \in \{0.75, 0.85, 0.90, 0.95, 0.99\}$ and uncertainty variance $\kappa_\tau(\sigma_0^u, \sigma_1^u, N_R) \in \{0.05, 0.08\}$. Table A.2 shows the above hyperparameters for `LEASE` under CQL and IQL algorithms. The maximum buffer capacity of unlabeled data is 50000. In next part, we give detailed analysis for the above hyper-parameters.

### B.3 PARAMETER ANALYSIS

**Number of preference data $N_l$.** The number of preference data $N_l$ influences the accuracy of reward model directly. `URLHF` (Yuan et al., 2024) trains reward model with 2000 preference dataset. Our method `LEASE` aims to achieve comparable performance with `URLHF` under fewer preference dataset. To test the effects of the number of labeled preference data for agent performance, we analyze the performance when the number of preference data $N_l$ is 20. The hyper-parameters of 20 preference dataset are same with that of 100 preference dataset, apart from the number of preference dataset and the batch size of pretrained reward model. The batch size is set as 16 here.

Table A.3: The comparison results for the D4RL tasks under different number of preference data $N_l$. $\pm$ captures the standard deviation over seeds. Bold indicates the highest score.

| CQL (Kumar et al., 2020) | | | IQL (Kostrikov et al., 2021b) | | |
|---|---|---|---|---|---|
| Task Name | $N_l = 20$ | $N_l = 100$ | Task Name | $N_l = 20$ | $N_l = 100$ |
| walker2d-medium | $77.7 \pm 1.3$ | $\mathbf{78.4 \pm 0.9}$ | pen-human | $71.9 \pm 7.4$ | $\mathbf{75.6 \pm 3.3}$ |
| walker2d-medium-expert | $98.0 \pm 18.6$ | $\mathbf{98.6 \pm 18.1}$ | pen-expert | $102.5 \pm 12.8$ | $\mathbf{113.8 \pm 6.3}$ |
| hopper-medium | $\mathbf{56.8 \pm 1.5}$ | $56.5 \pm 0.6$ | door-human | $4.4 \pm 1.2$ | $\mathbf{5.9 \pm 0.5}$ |
| hopper-medium-expert | $54.5 \pm 1.2$ | $\mathbf{56.4 \pm 0.8}$ | door-expert | $105.2 \pm 0.1$ | $\mathbf{105.2 \pm 0.2}$ |
| halfcheetah-medium | $43.4 \pm 0.4$ | $\mathbf{43.5 \pm 0.1}$ | hammer-human | $1.2 \pm 0.4$ | $\mathbf{1.7 \pm 0.4}$ |
| halfcheetah-medium-expert | $51.0 \pm 0.8$ | $\mathbf{53.2 \pm 3.1}$ | hammer-expert | $\mathbf{126.4 \pm 0.1}$ | $126.3 \pm 0.1$ |
| **Mujoco Average** | $63.6 \pm 4.0$ | $\mathbf{64.4 \pm 4.0}$ | **Adroit Average** | $68.6 \pm 3.7$ | $\mathbf{71.4 \pm 1.8}$ |

Table A.3 shows the agent performance when the number of preference dataset is 20 and 100, where we use CQL algorithm for mujoco tasks and IQL algorithm for adroit tasks. This table indicates `LEASE` still can improve agent performance under very fewer preference dataset. However, as the number of preference data $N_l$ decreases, the average of agent performance is slightly reduced. This is mainly because the reward model pretrained with very little preference data has a large generalization error, resulting in low quality of the generated preference data, which in turn affects

the accuracy of the final reward model. Notably, the performance gap between 100 preference data and 20 preference data is not much. This indicates that LEASE have potential to perform well under more fewer preference dataset.

**Rollout length $H$.** The rollout length $H$ is equal to the length of generated unlabeled data $\widehat{L}$. The $H$ is influenced by the accuracy of trained dynamics model. As $H$ increases, the accuracy of prediction decreases and large error will bring the unstable agent training. However, the long horizon of trajectory can better describe the preference of human feedback and can reduce sample complexity (Eq. (16)). Table A.4 gives the prediction accuracy of the trained reward model for different rollout length $H$. Here, we take CQL as example and select the last 200 of 2000 preference data as evaluation dataset that are not seen in in reward training stage.

The accuracy is calculated through the gap between predicted preference and true preference based on two trajectories. It shows that long horizon easily brings low performance of reward model since the cumulative error of the trained transition model, but long horizon is beneficial to enhance the performance of reward model under some conditions. The accuracy of preference model can be improved through data augmentation under most tasks. Note that the result can not completely represent the generalization performance of reward model since the evaluation data are limited. The choice of rollout length $H$ mainly depends on the accuracy of the trained transition model and the agent performance.

Table A.4: The prediction result of preference model for different rollout length $H$. The preference model is based on reward model. The performance of preference model can directly reflect the performance of reward model. The evaluation data are not seen in reward training stage.

| Task name | | walker2d-m | walker2d-m-e | hopper-m | hopper-m-e | halfcheetah-m | halfcheetah-m-e |
|---|---|---|---|---|---|---|---|
| **Pretrained model** | | **0.59** | 0.85 | 0.73 | 0.77 | 0.6 | 0.72 |
| **Updated** | $H = 10$ | 0.58 | **0.85** | **0.78** | 0.74 | **0.62** | **0.78** |
| | $H = 100$ | 0.57 | 0.84 | 0.71 | **0.79** | 0.57 | 0.74 |
| | $H = 200$ | 0.57 | 0.83 | 0.74 | 0.76 | 0.60 | 0.77 |

**Probability confidence $p$ and uncertainty variance $\tau$.** The above two parameters influence the selection of unlabeled preference dataset (Eq. (8)). For reward model composed of simple fully connected layers, using excessive amounts of unlabeled data for training can easily lead to over-fitting. Conversely, using too little data can result in poor generalization of the model. Implementing a selection mechanism ensures data quality on one hand while preventing the reward model from over-fitting on the other. Since the label is from the ground-truth reward for adroit tasks, the pre-trained reward model is closed to true model. Therefore, the probability confidence $p$ in adroit tasks is set higher than that of Mujoco tasks.

### B.4 ADDITIONAL RESULTS

**The other comparison results for screen mechanism.** Table A.5 shows the performance improvement under Adroit tasks when using screen mechanism $f(\sigma_0, \sigma_1)$. It shows that the performance can be significantly improved under CQL algorithm through selecting mechanism, but not significant under IQL algorithm. In general, combing with Table 2, we can conclude the selecting mechanism for unlabeled dataset can effectively improve the agent performance.

Table A.5: The comparison results between the performance using selecting mechanism and that not using. The latter method is denoted as FRESH. ↑ denotes the improvement of performance.

| Method | CQL (Kumar et al., 2020) | | | IQL (Kostrikov et al., 2021b) | | |
|---|---|---|---|---|---|---|
| | pen-expert | door-expert | hammer-expert | pen-expert | door-expert | hammer-expert |
| FRESH | $113.2 \pm 14.8$ | $101.6 \pm 3.5$ | $99.5 \pm 34.5$ | $113.4 \pm 14.7$ | $105.1 \pm 0.1$ | $125.5 \pm 0.8$ |
| LEASE | $\mathbf{132.5} \uparrow (\mathbf{17.0\%})$ | $\mathbf{103.2} \uparrow (\mathbf{1.6\%})$ | $\mathbf{126.3} \uparrow (\mathbf{26.5\%})$ | $113.8 \uparrow (0.3\%)$ | $105.2 \uparrow (0.2\%)$ | $126.3 \uparrow (0.7\%)$ |

**The other comparison results for reward model performance.** Fig. A.2 and A.3 show comparison between prediction value by the learned rewards and their ground truths for LEASE, FEWER and FRESH under other Mujoco and Adroit tasks, where the offline algorithm is IQL. The predicted and true rewards are both normalized to $[0, 1]$. We randomly sample 500 data from unlabeled datasets that are not seen in training stage for evaluation like Fig. 3. Since the reward has the value in

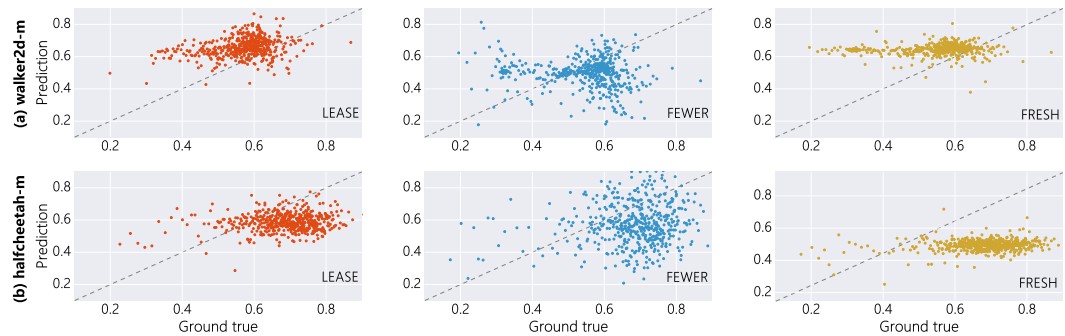

Figure A.2: The comparison between prediction value by the learned rewards and their ground truths for different methods under (a) walker2d-medium and (b) halfcheetah-medium datasets.

.

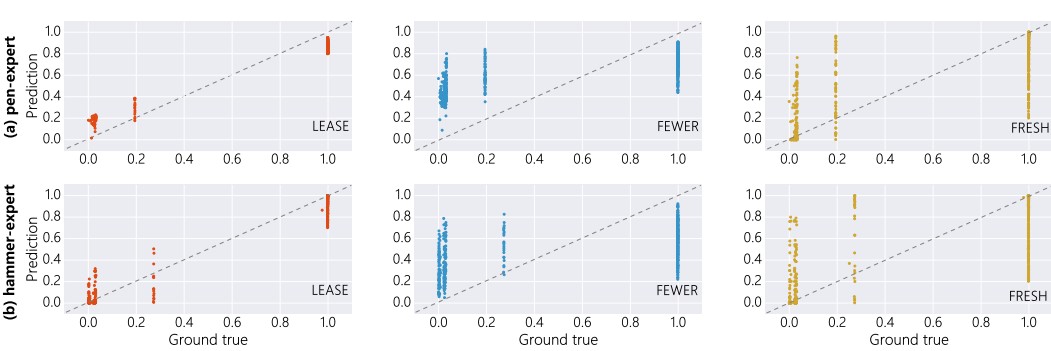

Figure A.3: The comparison between prediction value by the learned rewards and their ground truths for different methods under (a) pen-expert and (b) hammer-expert datasets.

.

certain position in Adroit tasks, the figure is in the form of multiple scattered lines in Fig. A.3. This also shows that the data augmentation and selecting mechanism can improve the reward model performance.

**Comparison results between `LEASE` and `URLHF` with fewer preference dataset.** To further show superior performance of the proposed method, Table A.6 compares `LEASE` to the baseline algorithm `URLHF` with the same amount of data as `LEASE`, where the latter method is denoted as URLHF*. This table shows that the average performance of `LEASE` is superior to that of the baseline algorithm `URLHF` using the same amount data with `LEASE`.

Table A.6: Comparison results between `URLHF` using fewer preference data and `LEASE`, where the offline RL algorithm is CQL and `URLHF` using fewer data is denoted as URLHF*.

| Task Name | URLHF* | URLHF | FEWER | LEASE |
|---|---|---|---|---|
| walker2d-m | $76.1 \pm 0.8$ | $76.0 \pm 0.9$ | $77.4 \pm 0.6$ | $\mathbf{78.4 \pm 0.9}$ |
| walker2d-m-e | $86.8 \pm 18.6$ | $92.8 \pm 22.4$ | $77.7 \pm 0.3$ | $\mathbf{98.6 \pm 18.1}$ |
| hopper-m | $\mathbf{56.6 \pm 2.4}$ | $54.7 \pm 3.4$ | $55.8 \pm 2.8$ | $56.5 \pm 0.6$ |
| hopper-m-e | $55.3 \pm 0.9$ | $\mathbf{57.4 \pm 4.9}$ | $53.6 \pm 0.9$ | $56.4 \pm 0.8$ |
| halfcheetah-m | $43.3 \pm 0.2$ | $\mathbf{43.4 \pm 0.1}$ | $43.5 \pm 0.1$ | $43.5 \pm 0.1$ |
| halfcheetah-m-e | $58.9 \pm 2.3$ | $\mathbf{62.7 \pm 7.1}$ | $48.3 \pm 0.7$ | $53.2 \pm 3.1$ |
| Mujoco Average | $62.8 \pm 4.2$ | $\mathbf{64.5 \pm 6.5}$ | $59.4 \pm 0.9$ | $64.4 \pm 4.0$ |
| pen-human | $\mathbf{17.7 \pm 13.0}$ | $9.8 \pm 14.1$ | $0.5 \pm 3.0$ | $3.8 \pm 4.6$ |
| pen-expert | $114.6 \pm 53.7$ | $\mathbf{138.3 \pm 5.2}$ | $128.1 \pm 0.7$ | $132.5 \pm 2.3$ |
| door-human | $1.7 \pm 1.1$ | $4.7 \pm 5.9$ | $0.2 \pm 1.0$ | $\mathbf{4.7 \pm 8.8}$ |
| door-expert | $103.3 \pm 0.5$ | $\mathbf{103.9 \pm 0.8}$ | $103.0 \pm 0.9$ | $103.2 \pm 0.7$ |
| hammer-human | $0.7 \pm 0.1$ | $\mathbf{0.9 \pm 0.3}$ | $0.3 \pm 0.0$ | $0.3 \pm 0.0$ |
| hammer-expert | $117.4 \pm 2.7$ | $120.2 \pm 6.8$ | $124.1 \pm 2.1$ | $\mathbf{126.3 \pm 1.2}$ |
| Adroit Average | $59.2 \pm 11.8$ | $\mathbf{63.0 \pm 5.5}$ | $59.4 \pm 1.3$ | $61.8 \pm 3.0$ |

**The effect of the introduction of uncertainty for pseudo-labeling accuracy.** Table A.7 validates the advantage of using uncertainty for reducing pseudo-labeling error, where the accuracy of pseudo-label generated by reward model is evaluated on all preference dataset. The below table shows that using uncertainty can improve accuracy of pseudo labels.

Table A.7: Comparison results of pseudo-labeling accuracy between using confidence and uncertainty and only using confidence.

| Task name | pen-expert | door-expert | hammer-expert |
|---|---|---|---|
| **confidence and uncertainty** | 87.25% | 89.25% | 85.45% |
| **only confidence** | 85.85% | 87.80% | 84.41% |

**The other results for LEASE under model-based offline algorithm.** Table A.8 shows the results of COMBO (Yu et al., 2021) under designed framework. For COMBO hyperparameters, the rollout horizon of preference trajectory $H$, probability confidence $\kappa_p$ and uncertainty variance $\kappa_\tau$ are set as 10, 0.85 and 0.05 for all tasks, respectively. Note that in our framework, model-based methods do not necessarily perform better than model-free methods. Model-based RL methods focus on how to learn conservative policy by regularizing $Q$ values or penalizing rewards to alleviate the effects of inaccuracy model data. Therefore, model-based RL requires higher accuracy of the reward model than model-free RL.

Table A.8: Comparison results of offline RL algorithms under the designed framework.

| Task Name | CQL$^*$ | IQL$^*$ | COMBO$^*$ |
|---|---|---|---|
| walker2d-m | $78.4 \pm 0.9$ | $74.6 \pm 1.8$ | $71.6 \pm 2.4$ |
| walker2d-m-e | $98.6 \pm 18.1$ | $108.1 \pm 0.5$ | $79.1 \pm 1.1$ |
| hopper-m | $56.5 \pm 0.6$ | $56.0 \pm 0.5$ | $54.8 \pm 0.9$ |
| hopper-m-e | $56.4 \pm 0.8$ | $55.9 \pm 1.9$ | $54.9 \pm 1.1$ |
| halfcheetah-m | $43.5 \pm 0.1$ | $43.0 \pm 0.3$ | $42.9 \pm 0.1$ |
| halfcheetah-m-e | $53.2 \pm 3.1$ | $62.4 \pm 1.4$ | $73.8 \pm 7.0$ |
| **Mujoco Average** | $64.4 \pm 4.0$ | $66.7 \pm 1.0$ | $62.9 \pm 2.1$ |

**The analysis of the accuracy of transition model for agent performance.** Table A.9 shows the detailed results for the effect of transition model accuracy for agent performance. It shows that the lower the accuracy of the transition model, the poorer performance of the agent, where the accuracy of transition model is measured by the sum of the mean square values of the predicted value and the true value in each dimension.

Table A.9: Results of the effect of the learned transition model for agent performance.

| hopper-medium | | pen-expert | |
|---|---|---|---|
| Transition model error | Agent performance | Transition model error | Agent performance |
| $0.35 \pm 0.01$ | $56.5 \pm 0.60$ | $1.05 \pm 0.01$ | $132.5 \pm 2.3$ |
| $0.49 \pm 0.04$ | $54.8 \pm 0.85$ | $1.42 \pm 0.05$ | $126.4 \pm 8.63$ |
| $1.19 \pm 0.05$ | $52.85 \pm 0.92$ | $2.36 \pm 0.24$ | $87.65 \pm 41.79$ |

## C  FURTHER DISCUSSION

### C.1  DISCUSSION FOR SAMPLE EFFICIENCY

**The discussion of sample efficiency.** In offline RL field, high sample efficiency refers that the agent can achieve comparable performance under fewer data compared with the performance under large data. In this paper, the data refers to the preference dataset. The labeled preference dataset, each trajectory of length $L$, is collected through real-time human feedback under policy $\mu$, which demands tremendous human effort, thus the collected cost of preference data is higher than fixed offline data. The unlabeled dateset is generated through trained transition without real-time interaction under learned policy $\pi^t$ at time $t$. Through data augmentation, the sample efficiency can be significantly reduced. The performance gap caused by reward can be reduced to $\varpi$ under fewer labeled dataset.

**Comparison with (Zhan et al., 2024).** Zhan et al. (2024) developed systematical theory for offline PbRL and also introduced the concentrability coefficient for PbRL, defined as

$$\mathscr{C}_r(\pi) = \max\left\{0, \sup_r \frac{\mathbb{E}_{\sigma^0 \sim \pi, \sigma^1 \sim \pi_{\text{ref}}}\big[r^*(\sigma^0, \sigma^1) - r(\sigma^0, \sigma^1)\big]}{\sqrt{\mathbb{E}_{\sigma^0 \sim \mu_0, \sigma^1 \sim \mu_1}\big|r^*(\sigma^0, \sigma^1) - r(\sigma^0, \sigma^1)\big|^2}}\right\}. \tag{A.26}$$

where $r(\sigma^0, \sigma^1) = \sum_i[R(s_i^0, a_i^0) - R(s_i^1, a_i^1)]$, $\pi_{\text{ref}}$ is an arbitrary trajectory distribution (usually set as $\mu_1$), and $\mu_0, \mu_1$ are behavior trajectory distribution. However, it is based on trajectory and is difficult to combine with other offline RL theories. The concentrability coefficient defined in Eq. (14) is based on state-action pairs.

Moreover, the performance gap of offline PbRL between behavior policy and learned policy is influenced by offline algorithm itself and the performance of the learned reward (preference) model. However, Zhan et al. (2024) fails to consider the gap caused by offline algorithm itself. The theory developed in our paper can be easily combined with other offline algorithm. The method in (Zhan et al., 2024) can learn $\varpi$-optimal policy with a sample complexity of

$$N = \widetilde{\mathcal{O}}\left(\frac{c^2 k^2 \mathscr{C}_r^2(\widehat{\pi})}{\varpi^2} \log\left(\frac{\mathcal{N}_r(1/N)}{\delta}\right)\right), \tag{A.27}$$

where $c > 0$ is a universal constant, $k = (\inf_{x \in [-r_{max}, r_{max}]} \Phi'(x))^{-1}$, and $\Phi(x)$ is a monotonically increasing link function. Compared with Eq. (A.27), the sample complexity of `LEASE` contains more useful information. For example, the sample complexity can be reduced when the length of preference data or the learned policy is closed to behavior policy.

## C.2 DISCUSSION FOR RELATED WORKS

**Model-free offline RL.** Existing model-free algorithms typically use two approaches: policy constraint and value regularization. The goal of policy constraint is to keep the learned policy close to the behavior policy (Kumar et al., 2019). On the other hand, value regularization methods mitigate the value overestimation for out-of-distribution (OOD) data by conservatively estimating $Q$ values (Kumar et al., 2020) or by penalizing based on the uncertainty of the $Q$ function (An et al., 2021). However, the core challenge for offline RL arises from limited data coverage. Model-free offline RL algorithms can only learn policies from the offline dataset, which restricts the agent's ability to explore.

**Model-based offline RL.** Model-based offline RL algorithms train a dynamics model using the offline dataset and leverage this model to enhance data coverage. However, due to the limitations of the offline dataset, there is a discrepancy between the learned dynamics model and the actual dynamics. To address this, conservatism should be integrated into the algorithms to prevent the agent from operating in areas where the predictions of the learned dynamics model are unreliable. One approach is to penalize the reward based on uncertainty quantification (Yu et al., 2020; Sun et al., 2023). Another approach is to enforce lower $Q$-values for data generated by the dynamics model that are deemed imprecise (Yu et al., 2021; Liu et al., 2023).

**Comparison with Surf (Park et al., 2022).** Surf is the PbRL method similar to `LEASE` using data augmentation technique. The differences between them mainly includes the below three aspects: 1) Surf belongs to online RL, but `LEASE` belongs to offline RL. Surf generates data through interaction with environment (simulator) while `LEASE` generates data through the learned transition model; 2) Surf only uses confidence for label filtering, whereas `LEASE` employs both confidence and uncertainty principles, which effectively reduce pseudo-label error; 3) `LEASE` provides the general theoretical framework for offline PbRL, but Surf don't provide theoretical analysis.

## C.3 BROADER IMPACTS

**Actual application.** In actual scenarios, especially for human-in-loop-control, such as exoskeleton robot assistance or rehabilitation, designing a high sample efficient RL algorithm is greatly significant. Firstly, many rewards are difficult to describe in mathematical terms in some situation, such as the comfort of interaction. Human feedback or preference is the better way to reflect the above indexes. In addition, in human in-the-loop control, obtaining preference data through interaction

can easily cause fatigue, and inappropriate interaction may cause damage to human. It is a necessity to learn reward model from limited preference dataset. Therefore, PbRL has a great potential to improve control performance in some scenarios where the reward function is difficult to describe and human is in the control loop.

**Theoretical study.** The theory of LEASE shows that the performance gap between the behavior policy $\mu$ and $\hat{\pi}$ learned by LEASE includes two part: the gap $\xi$ caused by offline algorithm itself and the gap $\xi_1$ caused by reward model gap (the details see Theorem 2), where the term $\xi_1$ only depends on preference dataset. Moreover, the theory of LEASE is based on state-action pairs, which is consistent with most offline algorithms theory. Therefore, the theory developed in this paper can be easily combined with other offline algorithms theory and easily used to build theory of policy improvement guarantee, which can provide the theoretical basis for offline PbRL and facilitate the further development of offline PbRL theory.

