# OpenReview forum: "LEASE: Offline Preference-based Reinforcement Learning with High Sample Efficiency"
_ICLR.cc/2025/Conference — ICLR 2025 Conference Withdrawn Submission_

### Official Review · Reviewer_T166 · 2024-10-21

**Soundness:** 3
**Presentation:** 3
**Contribution:** 3
**Rating:** 5
**Confidence:** 3

**Summary:**

This paper proposes a novel model-based offline RL algorithm that improves the efficiency of utilizing limited preference data. LEASE utilizes a learned transition model to rollout data, and label preferences with confidence and uncertainty measures. LEASE can achieve high performance with as few as 100 queries on mujoco tasks.

**Strengths:**

1. The problem of preference efficiency is vital and fundamental in offline meta-RL.
2. The proposed method is sound and sensible.
3. Experiment results are convincing.

**Weaknesses:**

1. Lack of analysis on the effect of transition model accuracy. The learned transition model can be inaccurate, and accumulate errors during rollout. Will this do a lot of damage to algorithm performance?
2. Lack of analysis of baseline algorithms' performance with different numbers of preference data. How much data is needed for baseline algorithms to achieve comparable performance to LEASE?
3. I would like to see results of more baseline algorithms, e.g., previous model-based offline RL algorithms.



Update after reviewer discussion:

After reading the other two Reviewers' reviews, I agree with them that experiments on D4RL are not that convincing, as previous works have shown that D4RL is largely insensitive to rewards. Considering this point, my major reason for raising the score, i.e., solid experiments, no longer remains convincing. Therefore, I agree with the other two reviewers that this paper is slightly below the acceptance threshold. I have modified my score accordingly.

**Questions:**

Please refer to the weakness part.

---

> ### Author Response · Authors · 2024-11-24
> **Response to Reviewer T166 (Part 1/2)**
>
> We deeply appreciate your thorough review and the valuable feedback provided on our work. Your questions indeed greatly improve overall quality of our work. Please refer to the updated PDF for new results and revisions.
>
> >**Question 1** : Lack of analysis on the effect of transition model accuracy. The learned transition model can be inaccurate, and accumulate errors during rollout. Will this do a lot of damage to algorithm performance?
>
> **Response 1**: Thank you for your valuable question. The accuracy of generated model data is the key factor influencing agent performance. We have provided the **detailed analysis for the effect of transition model accuracy for agent performance (See page 20, line 1218)** in the below table. It shows that the lower the accuracy of the transition model, the poorer performance of the agent, where the accuracy of transition model is measured by the sum of the mean square values ​​of the predicted value and the true value in each dimension.
> |hopper-medium||pen-expert||
> | :----:|:----:| :----:|:----:|
> |Transition Model Error |Agent Performance|Transition Model Error |Agent Performance|
> |$0.35\pm 0.01$|$56.5\pm0.60$| $1.05\pm0.01$|$132.5\pm2.3$|
> |$0.49\pm0.04$|$54.8\pm0.85$|$1.42\pm0.05$|$126.4\pm8.63$|
> |$1.19\pm0.05$|$52.85\pm0.92$|$2.36\pm0.24$|$87.65\pm41.79$|
>
> >**Question 2** : Lack of analysis of baseline algorithms' performance with different numbers of preference data. How much data is needed for baseline algorithms to achieve comparable performance to LEASE?
>
> **Response 2**: Thank you for your constructive suggestion. Table 1 in origin paper indicates that the performance of LEASE is superior to that of URLHF on some datasets. It is difficult to precisely determine the amount of data required for the baseline algorithm to achieve performance comparable to LEASE. Here, to further show superior performance of the proposed method, we compare LEASE to the baseline algorithm URLHF with the same amount of data as LEASE.
>
> We have compared the results between **LEASE and URLHF under the same number of data with LEASE (See page 22, line 1169)** in the below table, where the latter method is denoted as $\text{URLHF}^*$. This table shows that the average performance of LEASE is superior to that of the baseline algorithm URLHF using the same amount data with our method.
>
> |Task Name|$\textbf{URLHF}^*$|$\textbf{URLHF}$|$\textbf{FEWER}$|$\textbf{LEASE}$|
> |:----:|:----:| :----:|:----:|:----:|
> |walker2d-m | $76.1\pm0.8$|$76.0\pm0.9$ | $77.4\pm0.6$ | $\mathbf{78.4\pm0.9}$
> |walker2d-m-e| $86.8\pm18.6$| $92.8\pm22.4$ | $77.7\pm0.3$ | $\mathbf{98.6\pm 18.1}$ |
> |hopper-m| $\mathbf{56.6\pm2.4}$ | $54.7\pm3.4$ | $55.8\pm2.8$ | $\mathbf{56.5\pm0.6}$|
> |hopper-m-e| $55.3\pm0.9$| $\mathbf{57.4\pm4.9}$ | $53.6\pm0.9$ | $56.4\pm0.8$ |
> |halfcheetah-m|$43.3\pm0.2$ | $\mathbf{43.4\pm 0.1}$ | $\mathbf{43.5\pm0.1}$ | $\mathbf{43.5\pm0.1}$ |
> |halfcheetah-m-e| $58.9\pm2.3$|$\mathbf{62.7\pm7.1}$ | $48.3\pm0.7$ | $53.2\pm3.1$ |
> |**Mujoco Average**| $62.8\pm4.2$| $\mathbf{64.5\pm6.5}$ | $59.4\pm0.9$ | $\mathbf{64.4\pm4.0}$ |
> |pen-human | $\mathbf{17.7\pm13.0}$ | $9.8\pm14.1$ | $0.5\pm3.0$ | $3.8\pm4.6$ |
> |pen-expert| $114.6\pm53.7$| $\mathbf{138.3\pm5.2}$ | $128.1\pm0.7$ | $132.5\pm2.3$ |
> |door-human | $1.7\pm1.1$ | $\mathbf{4.7\pm5.9}$ | $0.2\pm1.0$ | $\mathbf{4.7\pm8.8}$ |
> |door-expert| $\mathbf{103.3\pm0.5}$| $\mathbf{103.9\pm0.8}$ | $103.0\pm0.9$| $\mathbf{103.2\pm 0.7}$ |
> |hammer-human | $0.7\pm0.1$| $\mathbf{0.9\pm0.3}$ | $0.3\pm0.0$ | $0.3\pm0.0$ |
> |hammer-expert| $117.4\pm2.7$| $120.2\pm6.8$ | $124.1\pm2.1$ | $\mathbf{126.3\pm1.2}$ |
> |**Adroit Average**| $59.2\pm11.8$| $\mathbf{63.0\pm5.5}$ | $59.4\pm1.3$ | $61.8\pm3.0$|

---

> > ### Author Response · Authors · 2024-11-24
> > **Response to Reviewer T166 (Part 2/2)**
> >
> > >**Question 3** : I would like to see results of more baseline algorithms, e.g., previous model-based offline RL algorithms.
> >
> > **Response 3**: Thank you for your valuable suggestion. In origin paper, CQL and IQL are both model-free offline RL algorithms. Applying the framework of the reward model we designed to model-based offline RL algorithms can further validate the effectiveness of our algorithm. Here, we choose popular model-based offline RL method: COMBO [1]. We have provided **model-based offline RL algorithm (COMBO) results under our designed framework (See page 23, line 1200)**. The results are given in the below table.
> >
> > |Task Name|$\textbf{CQL}^{\star}$|$\textbf{IQL}^{\star}$|$\textbf{COMBO}^{\star}$|
> > |:----:|:----:| :----:|:----:|
> > |walker2d-m |$78.4\pm 0.9$ | $74.6\pm1.8$|$71.6\pm2.4$ |
> > |walker2d-m-e|$98.6\pm 18.1$ | $108.1\pm0.5$|$79.1\pm1.1$ |
> > |hopper-m| $56.5\pm0.6$ |$56.0\pm0.5$ | $54.8\pm0.9$|
> > |hopper-m-e| $56.4\pm0.8$| $55.9\pm1.9$|$54.9\pm1.1$ |
> > |halfcheetah-m|$43.5\pm0.1$ | $43.0\pm0.3$| $42.9\pm0.1$|
> > |halfcheetah-m-e|$53.2\pm3.1$ | $62.4\pm1.4$| $73.8\pm7.0$|
> > |**Mujoco Average**|$64.4\pm 4.0$ |$66.7\pm1.0$ |$62.9\pm2.1$ |
> >
> > For COMBO hyperparameters, the rollout horizon of preference trajectory $H$, probability confidence $\kappa_p$ and uncertainty variance $\kappa_\tau$ are set as $10$, $0.85$ and $0.05$ for all tasks, respectively. The performance of COMBO under our framework may be further improved through optimize the above hyperparameters.
> >
> > Please note that in our framework, model-based methods do not necessarily perform better than model-free methods. Model-based RL methods focus on how to learn conservative policy by regularizing $Q$ values ​​or penalizing rewards to alleviate the effects of inaccuracy model data. Therefore, model-based RL requires higher accuracy of the reward model than model-free RL.
> >
> > **Reference**
> >
> > [1] Yu T, Kumar A, Rafailov R, et al. Combo: Conservative offline model-based policy optimization. *Advances in neural information processing systems*, 34, 2021.
> >
> > We are very sorry for the late response due to experimental reasons. Please check our corresponding response and revised PDF. If you have any questions, please feel free to ask us.

---

> > > ### Comment · Reviewer_T166 · 2024-11-25
> > > **Thank you for your thorough clarifiation**
> > >
> > > I thank the authors for providing a detailed and thorough clarification that addresses my concerns. I am raising my score accordingly.

---

> > > > ### Author Response · Authors · 2024-11-25
> > > > **Thank you for raising the score**
> > > >
> > > > We would like to thank the reviewer for raising the score to 8. We also appreciate the valuable comments and suggestions, which helped us significantly improve the overall quality of our paper!

---

### Official Review · Reviewer_CF9a · 2024-11-03

**Soundness:** 2
**Presentation:** 3
**Contribution:** 2
**Rating:** 5
**Confidence:** 3

**Summary:**

The paper studies the problem of improving sample efficiency in offline preference-based reinforcement learning (PbRL). The proposed LEASE algorithm aims to address this issue by generating synthetic unlabeled segment pairs using an ensemble of learned transition models. These synthetic pairs are subsequently labeled with an ensemble of pre-trained reward models, followed by a filtering process that ensures the quality of the pseudo labels. The filtering mechanism employs a confidence principle, which requires that the models have high certainty in discriminating between segment preferences, and an uncertainty principle, which stipulates low variance in the predictions of the ensemble models. An offline RL algorithm can then be employed to learn on the augmented labeled dataset. The paper also supports its claims with theoretical analysis, providing an upper bound on the reward model's error learned on an augmented dataset and a bound on the policy improvement. The empirical results on the D4RL benchmark demonstrate that LEASE achieves comparable performance to baselines while using less preference data.

**Strengths:**

1. The paper is well-organized, with the methodology and theoretical contributions presented clearly, making the core ideas easy to understand.
2. Improving sample efficiency in offline PbRL is an important and challenging problem, with significant implications for many real-world applications.
3. The paper provides contributions both in empirical algorithm development and theoretical analysis. Although there are concerns regarding the assumptions, the theoretical contributions add valuable insight into understanding the role of reward models and policy performance in PbRL.
4.  In addition to benchmark scores, the paper includes experiments that evaluate performance with varying amounts of preference data, as well as an analysis of the relationship between the learned reward model and the ground truth. This helps in understanding the effects of the different components of LEASE.

**Weaknesses:**

1. The proposed LEASE algorithm closely follows the pipeline of Surf[1], with only two major differences: (1) Surf augments data using random temporal cropping, whereas LEASE generates synthetic data with a learned transition model; (2) Surf only uses confidence for label filtering, whereas LEASE employs both confidence and uncertainty principles. Despite these differences, a direct comparison with Surf is missing, which would help clarify the effects of the introduced transition model and the uncertainty principle on the final performance. Without this comparison, it is challenging to establish the uniqueness or superiority of LEASE.
2. The main results in Table 1 compare LEASE with URLHF, a previous baseline that uses more data than LEASE. However, it is crucial to also include a comparison with URLHF using the same amount of data as LEASE. This would allow readers to determine whether LEASE truly achieves superior performance with fewer data or if the perceived gains are simply due to the different quantities of data used. The current results are not sufficiently convincing without this comparison.
3. The theoretical analysis relies on assumptions that may be unrealistic in practical settings, and the connection between theory and the empirical algorithm is weak. Specifically:
   - Assumption 2: Given a fixed learned reward model, it is possible to construct an adversarial unlabeled dataset such that the pseudo-labeling error \(\eta\) becomes very large. More detailed analysis is needed to understand whether the specific data generation process of LEASE can mitigate such worst-case scenarios effectively.
   - Filtering Mechanism: While it is intuitive that filtering low-quality data improves labeling accuracy, there is no clear theoretical justification for why the proposed filtering mechanism (confidence + uncertainty) is superior to previous methods that only use confidence.

[1] Jongjin Park, Younggyo Seo, Jinwoo Shin, Honglak Lee, Pieter Abbeel, and Kimin Lee. Surf:
Semi-supervised reward learning with data augmentation for feedback-efficient preference-based reinforcement learning. In 10th International Conference on Learning Representations, ICLR 2022.

**Questions:**

In addition to Weaknesses 1-3, we have following questions.
1. Some details of the algorithm are missing. In Line 6 of Algorithm 1, which policy is used together with the transition model to generate the data? In Line 8, the “reward update condition” is not clear. Additionally, there lacks an explanation for the “the reward model only update once instead of updating constantly in this process” statement.
2. Can Theorem 1 show that using augmented data is better? When $N_u=0$, it is clear that the constant term increases, but the pseudo-labeling error becomes zero. Is it possible that the bound is even tighter with no augmentation?
3. How well is the learned transition model? A bad transition model can lead to poor augmentation quality.
4. In Figure 3, “the linear relationship between reward predicted by LEASE and ground truth is better” is not very clear. What is the possible reason that FRESH’s predictions are very narrow?
5. Some minor problems:
(1) In the “Model-based Reinforcement Learning” part in section 2, model-based RL is not always offline, and the presentation is a bit.
(2) A ‘tilde’ is missing for $N_u$ in Equation 9.

---

> ### Author Response · Authors · 2024-11-24
> **Response to Reviewer CF9a (Part 1/3)**
>
> We would like to express our most sincere gratitude to you for your effort and patience in reviewing our paper. Your suggestions undoubtedly enhance the clarity, readability and overall quality of our work. Please refer to the updated PDF for new results and revisions.
>
> >**Question 1** : A direct comparison with Surf is missing, which would help clarify the effects of the introduced transition model and the uncertainty principle on the final performance.
>
> **Response 1**: We appreciate your valuable suggestion and patiently analyze the difference between our method and Surf. Apart from the differences you pointing, we **compared the differences between LEASE and Surf (See page 24, line1282)** in the below part.
> - **Surf belongs to online RL**. Surf generates unlabeled preference data through constantly interaction with environment (simulator). However, in some scenarios, it is a challenge to design a realistic simulator. If agent directly interacts with environment, it may bring dangers. Offline RL becomes a solution to this problem.
> - **LEASE belongs to offline RL**. The realistic simulator is not available for offline RL. To achieve data augmentation, LEASE trains the transition model to generate unlabeled preference data. Moreover, LEASE aims to achieve comparable performance under fewer preference data compared with baseline results under large amount of dataset.
> - **Theoretical contribution.** There are very fewer algorithms for offline PbRL theory. LEASE provides the general theoretical framework for offline PbRL, where the generalization bound of reward model and the theory of policy improvement are developed. The proposed theory can be easily combined with other offline RL algorithms.
>
> Please note that the agent performance using realistic simulator to augment data is superior to that using transition model intuitively. Response 7 also validates that the better transition model can improve agent performance. Therefore, we don't directly compare the performance between LEASE and Surf.
>
> Instead, we conducted experiments to **validate the advantage of using uncertainty for reducing pseudo-labeling error (See page 23, line 1188)**. We evaluated the accuracy of pseudo-label generated by reward model on all preference dataset. The below table shows that using uncertainty can improve accuracy of pseudo labels.
>
> |Task Name|pen-expert|door-expert|hammer-expert|
> |:----:|:----:| :----:|:----:|
> |confidence and uncertainty | $87.25\%$ | $89.25\%$ | $85.45\%$|
> |only confidence| $85.85\%$| $87.80\%$ | $84.41\%$ |
>
> **Reference**
>
> [1] Jongjin Park, et al. Surf: Semi-supervised reward learning with data augmentation for feedback-efficient preference-based reinforcement learning. In *10th International Conference on Learning Representations, ICLR 2022.*
>
> >**Question 2** : It is crucial to include a comparison with URLHF using the same amount of data as LEASE. This would allow readers to determine whether LEASE truly achieves superior performance with fewer data or if the perceived gains are simply due to the different quantities of data used.
>
> **Response 2**: Thank you for your valuable suggestion. Your suggestion provides a valuable opportunity to enhance the quality of our work. We have given the **comparison results between URLHF using fewer preference data and LEASE (See page 22, line1168)** in the below table, where RL algorithm is based on CQL and the method URLHF using fewer data is denoted as $\text{URLHF}^*$. It indicates that, in most tasks, LEASE can achieve superior performance compared with URLHF using the same amount of data as LEASE.
>
> |Task Name|$\textbf{URLHF}^*$|$\textbf{URLHF}$|$\textbf{FEWER}$|$\textbf{LEASE}$|
> |:----:|:----:| :----:|:----:|:----:|
> |walker2d-m | $76.1\pm0.8$|$76.0\pm0.9$ | $77.4\pm0.6$ | $\mathbf{78.4\pm0.9}$
> |walker2d-m-e| $86.8\pm18.6$| $92.8\pm22.4$ | $77.7\pm0.3$ | $\mathbf{98.6\pm 18.1}$ |
> |hopper-m| $\mathbf{56.6\pm2.4}$ | $54.7\pm3.4$ | $55.8\pm2.8$ | $\mathbf{56.5\pm0.6}$|
> |hopper-m-e| $55.3\pm0.9$| $\mathbf{57.4\pm4.9}$ | $53.6\pm0.9$ | $56.4\pm0.8$ |
> |halfcheetah-m|$43.3\pm0.2$ | $\mathbf{43.4\pm 0.1}$ | $\mathbf{43.5\pm0.1}$ | $\mathbf{43.5\pm0.1}$ |
> |halfcheetah-m-e| $58.9\pm2.3$|$\mathbf{62.7\pm7.1}$ | $48.3\pm0.7$ | $53.2\pm3.1$ |
> |**Mujoco Average**| $62.8\pm4.2$| $\mathbf{64.5\pm6.5}$ | $59.4\pm0.9$ | $\mathbf{64.4\pm4.0}$ |
> |pen-human | $\mathbf{17.7\pm13.0}$ | $9.8\pm14.1$ | $0.5\pm3.0$ | $3.8\pm4.6$ |
> |pen-expert| $114.6\pm53.7$| $\mathbf{138.3\pm5.2}$ | $128.1\pm0.7$ | $132.5\pm2.3$ |
> |door-human | $1.7\pm1.1$ | $\mathbf{4.7\pm5.9}$ | $0.2\pm1.0$ | $\mathbf{4.7\pm8.8}$ |
> |door-expert| $\mathbf{103.3\pm0.5}$| $\mathbf{103.9\pm0.8}$ | $103.0\pm0.9$| $\mathbf{103.2\pm 0.7}$ |
> |hammer-human | $0.7\pm0.1$| $\mathbf{0.9\pm0.3}$ | $0.3\pm0.0$ | $0.3\pm0.0$ |
> |hammer-expert| $117.4\pm2.7$| $120.2\pm6.8$ | $124.1\pm2.1$ | $\mathbf{126.3\pm1.2}$ |
> |**Adroit Average**| $59.2\pm11.8$| $\mathbf{63.0\pm5.5}$ | $59.4\pm1.3$ | $61.8\pm3.0$|

---

> > ### Author Response · Authors · 2024-11-24
> > **Response to Reviewer CF9a (Part 2/3)**
> >
> > >**Question 3** : Assumption 2: Given a fixed learned reward model, it is possible to construct an adversarial unlabeled dataset such that the pseudo-labeling error (\eta) becomes very large. More detailed analysis is needed to understand whether the specific data generation process of LEASE can mitigate such worst-case scenarios effectively.
> >
> > **Response 3**: Thank you for your constructive suggestion. Assumption 2 is made to enhance the universality of Theorem 1. Please note that the generalization error theory we proposed for the reward model is universal and applicable to methods that train reward model using pseudo-labels. The difference among various methods may lie in how they train reward model to improve the accuracy of pseudo-labels, that is how to reduce $\eta$. For our method, LEASE uses the selecting mechanism $f(\sigma_0,\sigma_1)$ to ensure the quality of pseudo-labels, which can reduce pseudo-labeling error $\eta$.
> >
> > **We have given more detailed analysis for the performance of reward model.** Assuming that the pretrained reward model has a label error rate of $\eta$. LEASE aims to use a screening mechanism $f(\sigma_0,\sigma_1)$ to select data with correct labels as much as possible. Then the reward model is updated with generated data to ensure that the newly reward model achieves a label error rate lower than $\eta$. However, if the performance of pretrained reward model is greatly poor, it would generate a large amount of wrong pseudo labels in beginning. This may result in the filtering mechanism being unable to guarantee the quality of generated labels, which leads to the collapse of the reward model training.
> >
> > >**Question 4** : Filtering Mechanism: While it is intuitive that filtering low-quality data improves labeling accuracy, there is no clear theoretical justification for why the proposed filtering mechanism (confidence + uncertainty) is superior to previous methods that only use confidence.
> >
> > **Response 4**: Thank you very much for pointing out the shortcomings of our paper. In Theorem 1, the filtering mechanism aims to reduce pseudo-label error $\eta$, and filtering mechanism has the effect on the term of pseudo-label error. Therefore, why the proposed filtering mechanism is better than the previous method using only confidence is equivalent to **why introducing uncertainty can reduce pseudo-label error.**
> >
> > Next, we explained why the introduction of uncertainty can reduce pseudo-label error from the perspective of ensemble model. The ensemble method is widely used to alleviate the inaccurate prediction of neural network. We estimated uncertainty through calculating the variance of ensemble model prediction. The previous works [1-2] have validated that the low uncertainty can improve prediction accuracy with high probability. Therefore, we selected pseudo-label with lower uncertainty, which can reduce the pseudo-label error. In response 1, we have conducted experiments to validate it.
> >
> > **Reference**
> >
> > [1] Liu J, Paisley J, Kioumourtzoglou M A, *et al.* Accurate uncertainty estimation and decomposition in ensemble learning. *Advances in neural information processing systems*, 32, 2019.
> >
> > [2] Lakshminarayanan B, Pritzel A, Blundell C. Simple and scalable predictive uncertainty estimation using deep ensembles. *Advances in neural information processing systems*, 30, 2017.
> >
> > >**Question 5** : Some details of the algorithm are missing. In Line 6 of Algorithm 1, which policy is used together with the transition model to generate the data? In Line 8, the “reward update condition” is not clear. Additionally, there lacks an explanation for the “the reward model only update once instead of updating constantly in this process” statement.
> >
> > **Response 5**: Thank you for your constructive suggestion. We are so sorry that we don't illustrate some details of algorithm clearly. We have give **more details for algorithm LEASE and explained why the reward model is only updated once (See page 6, line 321):**
> >
> > - In Line 6 of Algorithm 1, the current learned policy is used to generate data with transition model. In Line 8 of Algorithm 1, the reward would be updated when the number of preference data in buffer reaches the maximum buffer capacity. The update time is influenced by rollout frequency and rollout batch size.
> >
> > - Because the reward model is approximated by three-layer simple neural network, if the reward model is continuously updated during training, it may lead to overfitting, resulting in poor performance (we validate this in preliminary experiments). Therefore, the reward is only updated once during policy learning progress. Future work can further study reward model training adaptation mechanisms.

---

> > > ### Author Response · Authors · 2024-11-24
> > > **Response to Reviewer CF9a (Part 3/3)**
> > >
> > > >**Question 6** : Can Theorem 1 show that using augmented data is better? When $N_u=0$, it is clear that the constant term increases, but pseudo-labeling error becomes zero. Is it possible that the bound is even tighter with no augmentation?
> > >
> > > **Response 6**: Thank you for your constructive question. In the second-to-last inequality of Equation (A.2), we assume that the loss for each generated data with wrong label is at its maximum $\Omega$. This simplifies the theoretical form and directly reflects the factors affecting the error of the reward model. However, this leads to an issue of overestimation , which causes the form of bound is too loose in Theorem 1. In essential, the term pseudo-labeling error is lower than $\eta \Omega$.
> > >
> > > We have provided **more analysis for the generalization bound of reward model (See page 6, line 295).** Theorem 1 has generality and can be applicable to methods that train reward models using pseudo-labels. If one method fails to reduce the pseudo-labeling error $\eta$, the bound would be looser than no augmentation. However, for our method, we design selecting mechanism $f$ to reduce pseudo-labeling error $\eta$, and Figure 3 validates that the performance of reward model trained by LEASE is superior to that without no augmentation (FEWER), that is using augmentation for LEASE can improve the performance of reward model.
> > >
> > > >**Question 7** : How well is the learned transition model? A bad transition model can lead to poor augmentation quality.
> > >
> > > **Response 7**: Thank you for your valuable question. We have provided **the performance for the learned transition model (See page 9, line 478)** in the below table, where the accuracy of transition model is measured by the sum of the mean square values ​​of the predicted value and the true value in each dimension.
> > >
> > > Task Name|walker2d-m|walker2d-m-e|hopper-m|hopper-m-e|halfcheetah-m|halfcheetah-m-e|
> > > | :----:|:----:| :----:|:----:| :----:|:----:|:----:
> > > |Error|$10.19\pm 0.30$|$6.57\pm0.10$| $0.35\pm0.01$|$0.38\pm0.01$|$16.18\pm0.21$|$16.19\pm0.48$
> > > **Task Name**|**pen-human**|**pen-expert**|**door-human**|**door-expert**|**hammer-human**|**hammer-expert**|
> > > |Error|$1.02\pm 0.08$|$1.05\pm0.01$| $0.06\pm0.04$|$0.04\pm0.01$|$0.21\pm0.05$|$0.58\pm0.03$
> > >
> > > However, for walker2d and halfcheetah tasks, the error of the learned transition model is greatly larger than hopper tasks in mujoco environment. This is mainly because the state space of walker2d and hopper is larger than hopper. Walker2d and hopper contain more information of joint angle and angular velocity and the physical model of them are more complex.
> > >
> > > In addition, we provided the detailed analysis for **the effect of transition model accuracy for agent performance (See page 23, line 1218).** The below table shows that the lower the accuracy of the transition model, the poorer performance of the agent.
> > >
> > > |hopper-medium||pen-expert||
> > > | :----:|:----:| :----:|:----:|
> > > |Transition Model Error |Agent Performance|Transition Model Error |Agent Performance|
> > > |$0.35\pm 0.01$|$56.5\pm0.6$| $1.05\pm0.01$|$132.5\pm2.3$|
> > > |$0.49\pm0.04$|$54.8\pm0.85$|$1.42\pm0.05$|$126.4\pm8.63$|
> > > |$1.19\pm0.05$|$52.85\pm0.92$|$2.36\pm0.24$|$87.65\pm41.79$|
> > >
> > > >**Question 8** : In Figure 3, “the linear relationship between reward predicted by LEASE and ground truth is better” is not very clear. What is the possible reason that FRESH’s predictions are very narrow?
> > >
> > > **Response 8**: Thank you for your valuable question. Learning accurate reward model is still a challenging problem. The accuracy of reward model trained by LEASE indeed has a certain gap compared to the real model, but in the case of small samples, using our designed framework can effectively enhance the accuracy of the reward model's predictions (compared to FEWER and FRESH).
> > >
> > > We have given the **possible reason that FRESH’s predictions are very narrow (See page 9, line 468).** RRESH is the method where the generated unlabeled data are not screened through selecting mechanism $f(\sigma_0,\sigma_1)$. This may cause substantial errors for the labels of generated data. The introduction of more erroneous labels will lead to the collapse of reward model training, subsequently reducing prediction accuracy.
> > >
> > > >**Question 9** : Some minor problems: (1) In the “Model-based Reinforcement Learning” part in section 2, model-based RL is not always offline, and the presentation is a bit. (2) A ‘tilde’ is missing for $N_u$ in Equation 9.
> > >
> > > **Response 9**: Thank you for your careful and patient reading. **We have revised the subtitle of the second part (See page 3, line 143)** to Model-based Offline Reinforcement Learning. However, in Equation 9, the summation symbol should not have a tilde above the $N_u$, as $f(\sigma_0^u,\sigma_1^u)=0/1$ and $\sum_{i=0}^{N_u}f(\sigma_0^u,\sigma_1^u)=\tilde{N}_u$.
> > >
> > > We are very sorry for the late response due to experimental reasons. Please check our corresponding response and revised PDF. If you have any questions, please feel free to ask us.

---

> > > > ### Comment · Reviewer_CF9a · 2024-11-27
> > > > **Thank you for detailed response**
> > > >
> > > > I sincerely appreciate the authors for their detailed and thoughtful responses to my questions, as well as for the clarifications made in the paper. However, I still have major concerns that remain unresolved.
> > > >
> > > >
> > > > 1.Can LEASE outperform the baseline using the same amount of data?
> > > > In the results provided in Response 2, LEASE demonstrates only marginal improvements compared to URLHF*. In many tasks, the confidence intervals overlap, which raises doubts about the empirical superiority of LEASE when using limited offline data. Reviewer Sogr also pointed out, "D4RL benchmark is known to be insensitive to the accuracy of the reward function". But the two D4RL results in response 4 to reviewer Sogr may not provide sufficient evidence to support the claims made for a camera-ready paper.
> > > >
> > > >
> > > > 2.Connection to Surf and novelty of LEASE
> > > > The data augmentation and sample filtering techniques discussed in the paper are not inherently tied to whether the algorithm operates in an online or offline setting. LEASE remains conceptually similar to Surf. Although the authors, in Response 1, demonstrate that confidence and uncertainty filtering achieves slightly higher pseudo-label accuracy than Surf’s confidence-only filtering, the improvement is minimal. I am still wondering whether this narrow label accuracy gap will result in measurable improvements in the corresponding tasks.

---

> > > > > ### Author Response · Authors · 2024-11-29
> > > > > **Response to Reviewer CF9a (Part 1/2)**
> > > > >
> > > > > Thank you very much for taking the time to respond. We have answered your concerns in the below part.
> > > > >
> > > > > > **Question 1**: Can LEASE outperform the baseline using the same amount of data?
> > > > >
> > > > > **Response 1**:  Thank you for your valuable question. Please note that it is widely accepted that the current improved reinforcement learning algorithms struggle to perform well across all tasks, and the network architectures and parameters of URLHF and LEASE differ, which could also lead to the performance discrepancy. For the results of response 2, compared to URLHF*, LEASE shows significant improvements on several tasks, such as walker2d-m-e, pen-expert, and hammer-expert. For tasks like walker2d-m, hopper-m-e, halfcheetah-m, and door-human, the performance shows slight improvements.
> > > > >
> > > > > It can also be observed that in the hopper, halfcheetah, and door environments, the performance improvement of LEASE over URLHF* is minimal, and in some cases, even lower. This is primarily because the accuracy of the reward function in these environments does not have a significant impact on performance, which we have verified through experiments. The table below shows the performance comparison of the agent in the above three environments using official URLHF code, with different numbers of preference data (100, 500, 1000). This further validates the conclusion drawn above. In addition, the improvement in the accuracy of the preference model with an increase in the number of preference data has been verified in Figure 2 of the paper.
> > > > >
> > > > > |Task name|hopper-medium|  halfcheetah-medium| door-expert|
> > > > > |:----:|:----:| :----:|:----:|
> > > > > |100| $56.6\pm2.4$| $43.3\pm0.2$ |$103.3\pm0.5$ |
> > > > > |500| $55.8\pm0.1$| $43.4\pm0.2$ | $103.8\pm1.0$ |
> > > > > |1000| $55.6\pm2.2$| $43.3\pm0.2$ | $103.6\pm0.5$ |
> > > > >
> > > > > > **Question 2**: The two D4RL results in response 4 to reviewer Sogr may not provide sufficient evidence to support the claims made for a camera-ready paper.
> > > > >
> > > > > **Response 2**:  Thank you for your constructive comment. Regarding the benchmark experiments, the reviewer should note that D4RL and Meta-world are two different benchmarks. Our study mainly conducts experiments on two distinct domains (mujoco and adroit) within the D4RL benchmark, **covering 12 different tasks (6 locomotion tasks and 6 manipulation tasks)**. The baseline algorithm URLHF only provides experimental results for the D4RL benchmark.  In addition, not all tasks in the D4RL benchmark are insensitive to the accuracy of reward model. The comparison results between the URLHF and URLHF* also validates this.
> > > > >
> > > > > In response to Reviewer Sogr, the experiments in Response 4 involve two  tasks from Meta-world benchmark. However, our algorithm is designed for offline RL, and the D4RL benchmark provides publicly available offline datasets, whereas the Meta-world benchmark does not, requiring us to collect the data ourselves. Moreover, preference data for Meta-world is not provided by URLHF either. To maintain consistency and fairness in experimental comparisons, we did not include the results for the Meta-world benchmark in the revised PDF. Instead, we simply collected data and conducted a preliminary validation to demonstrate the effectiveness of our algorithm on the Meta-world benchmark.

---

> > > > > > ### Author Response · Authors · 2024-11-29
> > > > > > **Response to Reviewer CF9a (Part 2/2)**
> > > > > >
> > > > > > > **Question 3:** Connection to Surf and novelty of LEASE.
> > > > > >
> > > > > > **Response 3**:  Thank you for your valuable comment. From a technical perspective, both LEASE and Surf are preference-based reinforcement learning algorithms that enhance performance through data augmentation. However, **their motivations and application scenarios differ**. Surf focuses solely on the high cost of preference data collection, leveraging continuous interactions with the simulation environment to expand preference data and improve agent performance.
> > > > > >
> > > > > > In contrast, LEASE addresses multiple challenges: the high cost of preference data collection, the difficulty of designing simulation environments, and the risks of real-time online interaction. It aims to improve agent performance using a small amount of preference data. Surf relies on realistic simulation environments and is unsuitable for some human-in-the-loop control scenarios (It is challenging to simulate human subjective motor intentions). LEASE, on the other hand, has broader practical applications, as it can function by utilizing offline data alone.
> > > > > >
> > > > > > In addition, the proposed filtering mechanism is only part of the contribution of this work. The paper also includes **theoretical contributions**. Specifically, it provides a theoretical framework for offline preference-based reinforcement learning, analyzing the impact of the quality and quantity of preference data on the accuracy of the reward model, as well as the relationship between the agent's final performance, the offline algorithm itself, and the reward model. The reward model theory is applicable to reward models that use pseudo-labeling techniques, and the performance improvement theory can be readily integrated with other offline reinforcement learning algorithms.
> > > > > >
> > > > > > > **Question 4:** How much performance improvement is achieved after adding uncertainty?
> > > > > >
> > > > > > **Response 4**:  Thank you for your valuable question. The table below compares the accuracy of pseudo-labels and performance between our method and the confidence-only method. It can be concluded that our method outperforms the confidence-only approach in pseudo-labels accuracy and agent performance. The performance improved by 4.5%, 2.8% and 6.85% for pen-expert, door-expert and hammer-expert, respectively. Additionally, we kindly ask the reviewer to distinguish between the accuracy of the preference model $P(\sigma_0 \succ \sigma_1)$ (pseudo-label accuracy) and the accuracy of the reward model $R(s,a)$ (Please see equation 3 in revised PDF). A decline in the accuracy of the reward model may not necessarily lead to a significant drop in the accuracy of the preference model, as the preference model is related to the summation of reward model over trajectories.
> > > > > >
> > > > > > ||pen-expert| | door-expert| |hammer-expert||
> > > > > > |:----:|:----:| :----:|:----:|:----:|:----:|:----:|
> > > > > > ||label accuracy|agent performance|label accuracy|agent performance|label accuracy|agent performance|
> > > > > > |confidence and uncertainty | $87.3\pm0.5$ |$132.5\pm2.3$| $89.3\pm0.6$ | $103.2\pm 0.7$|$85.5\pm0.2$|$126.3\pm1.2$ |
> > > > > > |only confidence| $85.9\pm0.8$|$126.8\pm0.1$ |$87.8\pm0.2$ |$101.1\pm2.2$| $84.4\pm0.4$ | $118.2\pm9.6$|
> > > > > >
> > > > > > Thank you very much for your question. If you have any further questions, please feel free to ask us at any time.

---

> ### Comment · Reviewer_CF9a · 2024-12-01
>
> I would like to thank the authors for their detailed response.
>
> Regarding response 1 & 2, I believe it further substantiates Reviewer Sogr’s comment that the reward function in D4RL does not have a significant impact on performance, and D4RL may not be a suitable benchmark for the purpose of the paper. Specifically, LEASE demonstrates better performance in only 3 out of 12 D4RL tasks, while its performance in other tasks is either comparable or inferior. Furthermore, the average advantage observed is minimal relative to the variance. Other environments which are more sensitive to rewards (such as Meta-World) might provide more compelling evidence, though the current results remain insufficient to draw strong conclusions.
>
> On a positive note, I found the new results presented in Response 4 intriguing. They suggest that in tasks where LEASE does exhibit an advantage, there is a notable correlation between label accuracy and final performance.
>
> However, considering the overall scope and evidence provided in the paper’s current form, my evaluation remains unchanged.

---

> > ### Author Response · Authors · 2024-12-02
> >
> > Thank you very much for your response. Through our experiments, we have verified that in the hopper, halfcheetah, and door tasks, the accuracy of the reward has a limited impact on performance. This also explains why our method shows limited performance improvements on these tasks and, in some cases, even a decline. However, not all D4RL tasks exhibit insensitivity to reward accuracy. Reference [2] does not provide a corresponding conclusion either. Reference [2] primarily concludes that, on certain benchmark datasets, offline RL can achieve good performance even when trained with "incorrect" reward labels. This is mainly attributed to the pessimism in offline RL algorithms. Additionally, it observes that the Decision Transformer algorithm (DT) is mostly insensitive to reward quality (as stated on page 8 of Reference [2]).
> >
> > Conducting experiments on the Meta-world benchmark could indeed enhance the performance of the paper. However, all datasets and experiments in our study were initially based on Reference [1]. We also reproduced the performance of URLHF [1] on the Meta-world benchmark and found that training was difficult to stabilize (with preference labels generated from true reward labels). Thus, including these results in the paper for comparison is of limited significance. Moreover, the primary contributions of our work go beyond performance improvements. Specifically:
> > - This paper proposes a novel offline preference-based reinforcement learning framework, demonstrating how to achieve comparable performance with a limited amount of preference data.
> > - The theories about reward model and performance improvement are derived, where the reward model theory is applicable to reward models that use pseudo-labeling techniques, and the performance improvement theory can be readily integrated with other offline RL algorithms.
> >
> > Once again, thank you very much for your response.
> >
> > **Reference**
> >
> > [1] Yifu Yuan, et al. "Uni-rlhf: Universal platform and benchmark suite for reinforcement learning with diverse human feedback." In *12th International Conference on Learning Representations*, 2024.
> >
> > [2] Li, Anqi, et al. "Survival instinct in offline reinforcement learning." *Advances in neural information processing systems*, 2024.

---

### Official Review · Reviewer_Sogr · 2024-11-04

**Soundness:** 2
**Presentation:** 2
**Contribution:** 2
**Rating:** 5
**Confidence:** 4

**Summary:**

The paper presents a novel algorithm for offline preference-based reinforcement learning (PbRL) aimed at addressing the challenges of designing rewards and the high costs of online interaction. The LEASE algorithm leverages a learned transition model to generate unlabeled preference data, which is then filtered through an uncertainty-aware mechanism to ensure the performance of the reward model. The paper claims to provide a generalization bound for the reward model and a theoretical improvement guarantee for the policy learned by LEASE. The experimental results are said to demonstrate that LEASE can achieve comparable performance to baseline methods with fewer preference data without online interaction.

**Strengths:**

- The proposed LEASE algorithm is novel, which introduces a new way to handle the challenge of limited preference data with a learned transition model. The selection mechanism for unlabeled data based on confidence and uncertainty is a thoughtful contribution to improving the stability and accuracy of the reward model.

-Theoretical Framework: The paper attempts to provide a theoretical foundation for the algorithm, which is a step towards more principled offline PbRL methods.

**Weaknesses:**

- While the paper provides a theoretical analysis of the algorithm, it is not rigorous enough. For example, the approximation error of the reward function usually depends on a condition number that is exponential to $R_{\text{max}}$ when learned from preference data [1], but it seems missing from the derived bound. The approximation error of reward error does not directly translate into an additional error term in the performance bound and requires careful treatment (e.g., use a pessimistic reward function [2]). The authors use a handwaving argument (i.e., the law of large numbers) to derive (A.24) from (A.23), but it is not accurate. Using a concentration inequality is necessary in the finite sample case.

- The paper lacks some benchmarks and baselines to validate the effectiveness of the proposed method. For benchmarks, The D4RL benchmark is known to be insensitive to the accuracy of the reward function [3], and adding benchmarks like Meta-World would greatly strengthen the paper. Also, there are some recent works on offline PbRL that have a strong performance like [4,5], and LEASE should be compared with them.


References

[1] Pacchiano, Aldo, Aadirupa Saha, and Jonathan Lee. "Dueling rl: reinforcement learning with trajectory preferences." arXiv preprint arXiv:2111.04850 (2021).

[2] Hu, Hao, et al. "The provable benefits of unsupervised data sharing for offline reinforcement learning." arXiv preprint arXiv:2302.13493 (2023).

[3] Li, Anqi, et al. "Survival instinct in offline reinforcement learning." Advances in neural information processing systems 36 (2024).

[4] Kim, Changyeon, et al. "Preference transformer: Modeling human preferences using transformers for rl." arXiv preprint arXiv:2303.00957 (2023).

[5] Zhang, Zhilong, et al. "Flow to better: Offline preference-based reinforcement learning via preferred trajectory generation." The Twelfth International Conference on Learning Representations. 2023.

**Questions:**

See the Weakness section.

---

> ### Author Response · Authors · 2024-11-24
> **Response to Reviewer Sogr (Part 1/2)**
>
> Thank you for your kind and precious comments. It is our honor that you can review our research. Your questions provide a valuable opportunity for us to improve the clarity, readiness and quality of our article. Please refer to the updated PDF for new results and revisions.
>
> >**Question 1**: The approximation error of the reward function usually depends on a condition number that is exponential to $R_{max}$ when learned from preference data [1], but it seems missing from the derived bound.
>
> **Response 1**: Thank you for your valuable comment. Paper [1] mainly derives the regret bound of performance, but does not analyze the bounds of the reward model in detail. The generalization bound of reward model (Theorem 1) for our method is based on statistics machine learning theory. Theorem 1 has generality and can be applicable to methods that train reward model using pseudo-labels. The difference among various methods may lie in how they train reward model to improve the accuracy of pseudo-labels, that is reducing $\eta$.
>
> In Theorem 1, the Empirical Rademacher Complexity (ERC) is used to measure the space complexity of the reward model space $\mathcal{R}$. Assume that the reward model is controlled by a parameter matrix $H$ (the weight matrix of the neural network). If the condition number of $H$ is too large, it may be assumed that there are overly complex or ill-conditioned functions in the space, resulting in an increase in ERC. Therefore, the ERC itself can reflect some characteristic of the reward model function.
>
> >**Question 2**: The approximation error of reward error does not directly translate into an additional error term in the performance bound and requires careful treatment (e.g., use a pessimistic reward function [2]).
>
> **Response 2**: Thank you for your valuable comment. In Eq. (A.12), we theoretically derive that the performance bound includes two parts: the offline RL algorithm related term and the reward model related term. The performance bound is derived from a rigorous form including the reward model error term (See Eq. A.14). **For paper [2], it improves the offline PbRL algorithm itself** (Algorithm 1 in [2]), and aims to ensure the conservatism of the algorithm through adding additional penalties on the reward function to prevent overestimation.
>
> The final form of the theory is related to the algorithm itself, **so the pessimistic reward function is used in the final performance bound of [2]**. However, our algorithm provided the universal theoretical framework of offline PbRL and did not improve the offline PbRL algorithm itself, so the final performance bound did not include other term like the pessimistic reward function. Future work can focus on **how to achieve conservative estimation for state-action pairs where the learned reward model predicts inaccurately (See page 10, line 521)**.
>
>
> >**Question 3** : The authors use a handwaving argument (i.e., the law of large numbers) to derive (A.24) from (A.23), but it is not accurate. Using a concentration inequality is necessary in the finite sample case.
>
> **Response 3**: Thank you for your valuable suggestion. Your suggestion indeed enhance the theoretical rigor of paper. In the finite sample case, concentration inequalities are crucial for providing precise bounds on the error between the sample mean and the expected value. According to *Chernoff-Hoeffding* bound [7], for independent random variables $X_1,X_2,...,X_n$, with high probability $1-\delta$, the below equation holds
> $$
> \mathbb{E}[X]\leq \frac{1}{n}\sum_{i=1}^{n}X_i +(b-a)\sqrt{\frac{\log (1/\delta)}{2n}},
> $$
> where $[a,b]$ is the range of values that each $X_i$ can take. Then, for equation (A.23), $X_i=R^*(s_j,a_j)-R(s_j,a_j)$ and $X_i \in[-2R_{max},2R_{max}])$. Therefore, equation (A.24) should be added the below term
>
> $$\sqrt{\frac{4R_{max}^2\log (1/\delta)}{NL}}.$$
>
> Thank you again for pointing out our problem! We have **revised some related equations in origin paper (See page 18, line 938; page 7, line 348-360).**

---

> > ### Author Response · Authors · 2024-11-24
> > **Response to Reviewer Sogr (Part 2/2)**
> >
> > >**Question 4** : The D4RL benchmark is known to be insensitive to the accuracy of the reward function [3], and adding benchmarks like Meta-World would greatly strengthen the paper.
> >
> > **Response 4**: Thank you for your constructive suggestion. The dataset of our algorithm is based on the paper [6]. LEASE aims to achieve control performance comparable to that of the paper [6] using a small amount of preference data. We verified the performance of the algorithm on two types of tasks: **locomotion tasks (gym-mujoco) and manipulation tasks (adriot)**. LEASE belongs to offline RL, and D4RL benchmark has a dedicated offline dataset while Meta-World does not have. The paper [6] also doesn't provide the preference data of Meta-World.
> >
> > To test the performance on the Meta-World benchmark, following the previous work [8], we used the trained policy to collect offline data and used real rewards to generate labels for the preference data. The below table shows the performance of our framework on two tasks of Meta-World benchmark. The offline algorithm is based on IQL. This also preliminarily shows the advantages of our method under this benchmark.
> >
> > |Task Name|drawer-open-v2|sweep-into-v2|
> > |:----:|:----:| :----:|
> > |URLHF (2000) |$18.7\pm 5.5$ | $85.3\pm3.1$|
> > |LEASE (100)|$34.0\pm 8.5$ | $96.7\pm1.2$|
> >
> > >**Question 5** : There are some recent works on offline PbRL that have a strong performance like [4,5], and LEASE should be compared with them.
> >
> > **Response 5**: Thank you for your valuable comment. Comparing with the most recent cutting-edge methods can indeed improve the quality of the article. However, the purpose of LEASE is to improve sample efficiency and achieve better performance with a small amount of data. We did not compare with [4-5] mainly due to the following two aspects:
> > - **Different preference data:** The starting point of LEASE is based on the article [6], how to achieve comparable performance with fewer preference data. The preference data of LEASE is based on the preference data provided by URLHF, which is different from the preference data of [4-5].
> > - **Different network complexity:** LEASE aims to provide a universal framework, all networks are approximated by simple neural networks, and can be easily combined with other offline RL algorithms. However, [4] and [5] use complex networks (transformer and diffusion model), and increase training costs to improve performance.
> >
> > Future research can be conducted to test the performance of the proposed framework using complex networks and compare the recent offline PbRL methods based on same preference dataset.
> >
> > **References**
> >
> > [1] Pacchiano, Aldo, Aadirupa Saha, and Jonathan Lee. "Dueling rl: reinforcement learning with trajectory preferences." arXiv preprint arXiv:2111.04850, 2021.
> >
> > [2] Hu, Hao, et al. "The provable benefits of unsupervised data sharing for offline reinforcement learning." arXiv preprint arXiv:2302.13493, 2023.
> >
> > [3] Li, Anqi, et al. "Survival instinct in offline reinforcement learning." *Advances in neural information processing systems* 36, 2024.
> >
> > [4] Kim, Changyeon, et al. "Preference transformer: Modeling human preferences using transformers for rl." arXiv preprint arXiv:2303.00957, 2023.
> >
> > [5] Zhang, Zhilong, et al. "Flow to better: Offline preference-based reinforcement learning via preferred trajectory generation." in *11th International Conference on Learning Representations*. 2023.
> >
> > [6] Yifu Yuan, et al. "Uni-rlhf: Universal platform and benchmark suite for reinforcement learning with diverse human feedback." In *12th International Conference on Learning Representations*, 2024.
> >
> > [7] Wassily Hoe ding. Probability inequalities for sums of bounded random variables. *Journal of the American Statistical Association*, 58(301):1330, 1963.
> >
> > [8] Choi H, et al. "Listwise reward estimation for offline preference-based reinforcement learning". arXiv preprint arXiv:2408.04190, 2024.
> >
> > We are very sorry for the late response due to experimental reasons. Please check our corresponding response and revised PDF. If you have any questions, please feel free to ask us.

---

### Author Response · Authors · 2024-11-24
**General Response**

We would like to express our sincere gratitude to the reviewers for valuable comments and suggestions that have greatly helped us improve the overall quality of our work. We are greatly encouraged by the positive comments of reviewers, e.g.,
- The proposed method is novel and sound, which provides contributions both in theoretical analysis and empirical algorithm development.
- Theory: This paper provides a theoretical foundation for the proposed algorithm, which is a step towards more principled offline PbRL methods.
- Experiment: In addition to evaluating locomotion and manipulation tasks, this paper conducts experiments to analyze the effects of different components of LEASE.

We have incorporated the reviewers' suggestions by adding theoretical explanation, experimental analysis, and discussions on related works. The significant revisions are summarized as follows:
- **Theoretical aspects**: We explained the generalization bound of reward contains more reward information(reviewer Sogr, response 1) and why translating reward model error term into performance bound (reviewer Sogr, response 2), revised the imprecise derivations (reviewer Sogr, response 3), analyzed the effect of pseudo-label error (Assumption 2) for reward bound (Theorem 1) in detail (reviewer CF9a, responses 3 and 6), and explained the screening mechanism how to reduce pseudo-label error (reviewer CF9a, response 4).
- **Experimental aspects**: We conducted experiments to analyze the accuracy of the learned transition model (reviewer CF9a, response 7) and the effect of it for agent performance (reviewer T166, response 1), compared the result between LEASE and baseline algorithm under the same amount of data with LEASE (reviewer CF9a, response 2 and reviewer T166, response 2),  and validated the performance of designed framework for meta-world benchmark (reviewer Sogr, response 4) and model-based offline method (reviewer T166, response 3).
- **Analysis aspects**: We provided the differences between LEASE and the related work Surf (reviewer CF9a, response 1), explained why we don't directly compare the recent related methods (reviewer Sogr, response 5), analyzed the reason why the reward prediction of FRESH is narrow (reviewer CF9a, response 8) and the advantages of introducing uncertainty (reviewer CF9a, response 1).

We have incorporated new analysis and experimental results and made revisions to the PDF in response to the reviewers' suggestions. We kindly invite you to review the updated version of our paper, where the changes have been highlighted for your convenience.

---

### Note · Authors · 2024-12-30

I have read and agree with the venue's withdrawal policy on behalf of myself and my co-authors.